# pGlycoQuant with a deep residual network for quantitative glycoproteomics at intact glycopeptide level

Siyuan Kong[1,6], Pengyun Gong [2,6], Wen-Feng Zeng [3,4,6], Biyun Jiang[1,6], Xinhang Hou[2], Yang Zhang[1], Huanhuan Zhao[1], Mingqi Liu [1], Guoquan Yan[1], Xinwen Zhou[1], Xihua Qiao[2], Mengxi Wu[1], Pengyuan Yang [1,5,7], Chao Liu [2,3] ✉ & Weiqian Cao [1,5] ✉

Large-scale intact glycopeptide identification has been advanced by software tools. However, tools for quantitative analysis remain lagging behind, which hinders exploring the differential site-specific glycosylation. Here, we report pGlycoQuant, a generic tool for both primary and tandem mass spectrometry-based intact glycopeptide quantitation. pGlycoQuant advances in glycopeptide matching through applying a deep learning model that reduces missing values by 19–89% compared with Byologic, MSFragger-Glyco, Skyline, and Proteome Discoverer, as well as a Match In Run algorithm for more glycopeptide coverage, greatly expanding the quantitative function of several widely used search engines, including pGlyco 2.0, pGlyco3, Byonic and MSFragger-Glyco. Further application of pGlycoQuant to the N-glycoproteomic study in three different metastatic HCC cell lines quantifies 6435 intact N-glycopeptides and, together with in vitro molecular biology experiments, illustrates site 979-core fucosylation of L1CAM as a potential regulator of HCC metastasis. We expected further applications of the freely available pGlycoQuant in glycoproteomic studies.

Protein glycosylation has long been known as a heterogeneous post-translational modification (PTM) that increases protein diversity and exerts a profound effect on various biological processes[1–4]. In spite of some issues remaining unsolved[5], great strides have been made in the identification of intact glycopeptides on a proteome-wide scale with the development of mass spectrometry (MS)-based analytical methods and interpretation software tools[6–14].

Growing implications of glycosylation in physiological and pathological processes have prompted an intensive focus on studying altered site-specific glycans through quantitative glycoproteomics[14–17]. The primary and tandem mass spectrometry (MS1/MS2)-based

quantitative strategies, such as label-free, isotope chemical labeling and isotope metabolic labeling approaches, have been accepted as gold standard methods for proteomics quantitative analysis[18,19]. The wide application of these strategies in large-scale quantitative intact glycoproteomic studies has been impeded by the lack of mature software tools for quantitative data processing[7,20]. Although some recently developed strategies are suitable for intact glycopeptide quantitation[11,12,21,22], many issues still remain. For example, intact glycopeptide quantitation suffers from impaired accuracy and large numbers of quantitative missing values since glycopeptide signals are more difficult to recognize than that of naked peptides due to the

[1]Shanghai Fifth People's Hospital and Institutes of Biomedical Sciences, Fudan University, Shanghai, China. [2]School of Engineering Medicine & School of Biological Science and Medical Engineering, Beihang University, Beijing, China. [3]Key Lab of Intelligent Information Processing of Chinese Academy of Sciences (CAS), Institute of Computing Technology, CAS, Beijing, China. [4]Proteomics and Signal Transduction, Max Planck Institute of Biochemistry, Martinsried, Germany. [5]NHC Key Laboratory of Glycoconjugates Research, Fudan University, Shanghai, China. [6]These authors contributed equally: Siyuan Kong, Pengyun Gong, Wen-Feng Zeng, Biyun Jiang. [7]Deceased: Pengyuan Yang. ✉e-mail: liuchaobuaa@buaa.edu.cn; wqcao@fudan.edu.cn

microheterogeneity. In addition, the data dependent acquisition (DDA) strategy-based deep quantitative glycoproteome coverage is limited to the MS2 level for the reason that it is normal to not identify all the glycoforms present due to them not all being selected for MS/MS analysis. Efficient software tools for the reliable and global quantitative glycoproteomic analysis at intact glycopeptides level are greatly needed[7,16,20].

A targeted mass spectrometry signal is easily affected by interference from nearby signals or noise, and its morphological characteristics cannot be entirely remained, resulting in impaired accuracy or missing values[23]. Deep learning-based algorithms have led to very good performance on a variety of subjects[24,25]. Among them, the deep residual neural network (ResNet) has been accepted as an effective method for training computational vision object detection models that can represent much more complex functions than were previously practically feasible[26]. The main benefit of ResNet is that an image or matrix could be transformed to a well-trained vector that shows excellent performance in learning patterns from complex data and in matching two matrices[27,28].

Here, we present pGlycoQuant, a dedicated software tool for large-scale and global quantitative glycoproteomics. pGlycoQuant advances in glycopeptide evidence matching through applying a deep learning model that improves Match Between Run (MBR) performance, as well as an optional function of Match In Run (MIR) algorithm for more quantitative coverage of glycopeptides. We applied pGlycoQuant to state-of-the-art glycopeptide quantification analysis and comparison with other quantitation software tools, including MSFragger-Glyco, Byologic™, Skyline, and Proteome Discoverer. pGlycoQuant reduces missing values for glycopeptide quantification by 19–89% compared with other quantitative software tools. The current version of pGlycoQuant supports both primary and tandem mass spectrometry quantitation for multiple quantitative strategies, including label-free, chemical labeling and metabolic labeling approaches, and is compatible with identification results from several widely used search engines, including the Byonic[29], MSFragger-Glyco[12], Open-pFind[30], and pGlyco series engines[9,13]. Furthermore, a pGlycoQuant-based site-specific N-glycoproteomic study quantified 6435 intact N-glycopeptides in three hepatocellular carcinoma (HCC) cell lines with different metastatic potentials and, together with in vitro molecular biology experiments, identified core fucosylation at site 979 of the L1 cell adhesion molecule (L1CAM) as a potential regulator of HCC metastasis.

## Results

### Development and optimization of pGlycoQuant
Intact glycopeptide signals are difficult to recognize due to the microheterogeneity of glycosylation and some low-abundance signals (Fig. 1a), which may result in impaired accuracy and large quantitative missing values. Here, we developed pGlycoQuant, a dedicated software tool, for large-scale and global glycoproteomic quantitation at intact glycopeptide level. Currently, pGlycoQuant supports both primary and tandem mass spectrometry quantitation for multiple quantitative strategies, including label-free, chemical labeling and metabolic labeling approaches, and is compatible with several widely used search engines, including Byonic[29], MSFragger-Glyco[12], Open-pFind[30] and pGlyco series[9,12] (Fig. 1b). The workflow of pGlycoQuant consists of three steps: identification result reading, signal extracting, and quantitation processing (Supplementary Fig. 1, detailed in the "Methods" section). In the quantitation processing step, it applies the ResNet deep learning to the glycopeptide evidence matching model for fine glycopeptide matching and MBR analysis (Fig. 1c, Supplementary Fig. 2, also see the "Methods" section) and includes a false quantitation rate (FQR) estimation method for ruling out false quantitative results of MBR analysis with 1% FQR (Fig. 1d and Supplementary Fig. 3). In addition, an optional function of MIR was proposed for increasing quantitative coverage of glycopeptides (Fig. 1e).

In the deep learning-based evidence matching model (Supplementary Fig. 2), a glycopeptide evidence is first mapped to a tensor of $512 \times 1$, and the best signal patterns are retained by the ResNet18 model. Then, a fully connected network that comprehensively utilizes multiple characteristics, including the similarity of isotopic peaks and distance of retention time between glycopeptides, is trained to measure the similarity of two glycopeptide evidences in the same run for metabolic-labeling data or between different runs for label-free data. Finally, the model provides the matching score for each quantitation result as a softmax loss function is used in the network (detailed in the "Methods" section). In the MBR analysis, the matching scores were further used to calculate the FQR by a target-decoy Gaussian mixture model approach fitted with Expectation-Maximization (EM) algorithm for quantitative result control with 1% FQR (Fig. 1c and Supplementary Fig. 3). Details of FQR estimation approach are described in the "Methods" section.

### Comprehensive and comparative evaluation of pGlycoQuant
First of all, we evaluated the quantitative accuracy of pGlyco-Quant with MBR analysis through two dedicated experimental designs, a two-glycoproteome experiment and a fold change-(de)glycoproteome experiment. In the two-glycoproteome experiment, we performed intact glycopeptide quantitation with MBR in yeast and human samples (Supplementary Fig. 4a, Supplementary Note 1). Since the glycopeptides in the two samples are different, it is theoretically impossible to quantify any glycopeptides from one sample to the other. Therefore, a successful quantitation of a glycopeptide from one sample to the other sample leads to a false positive quantitation. The ratio of false positives was calculated as the FQR-entrapment, which was used to estimate the validity of glycopeptide quantitation by pGlycoQuant with MBR analysis (Supplementary Fig. 4a). The results showed that the entrapment-based FQR for the GPSMs (glycopeptide spectra matches) and glycopeptides reported by pGlycoQuant were 0.36% and 0.88%, respectively, which were below the preset criterion of a 1% FQR (Supplementary Fig. 4b). In the fold change-(de)glycoproteome experiment, we performed experiments that compare the same sample after glycopeptide enrichment with four conditions (Supplementary Fig. 5a and Supplementary Note 2).The results showed that the known ratio was well recovered by pGlycoQuant for the 5-fold change glycopeptides (Supplementary 5b–d). Meanwhile, rare glycopeptides could be quantified in deglycosylation samples (<0.8% for fission yeast samples and <1.6% for human serum samples) after MBR analysis between glycopeptide and deglycopeptide samples (Supplementary 5e–g). The above analyses demonstrated that pGlycoQuant could reliably quantify glycopeptide evidences instead of other interference peaks.

Then, we compared pGlycoQuant with prevalently used search engines that equipped with quantitative functions, namely Byonic-Byologic and MSFragger-Glyco, as well as the quantitation software Skyline and Proteome Discoverer, for intact glycopeptide quantitation on three benchmark datasets, including SILAC-labeled 293 T cell data, label-free HeLa cell data, and TMT-labeled 293T cell data (Fig. 2a and Supplementary Tables 1–3).

The low quantitative reproducibility and consistency is often manifested as data missing values. We defined two indicators, the proportion of missing values in line (PMVL) and the proportion of missing values in total (PMVT), to measure the proportion of missing values (Supplementary Fig. 6). In label-free quantitation, 61.45% PMVL and 30.12% PMVT at glycopeptide level were reported by Byologic (Fig. 2b and Supplementary Table 4). Although the missing value problem was ameliorated in Proteome Discoverer, MSFragger-Glyco and Skyline, there were still 10.76% PMVL and 4.89% PMVT, 27.68% PMVL and 13.45% PMVT, and 18.19% PMVL and 17.81% PMVT reported by the three software tools, respectively (Fig. 2b and Supplementary

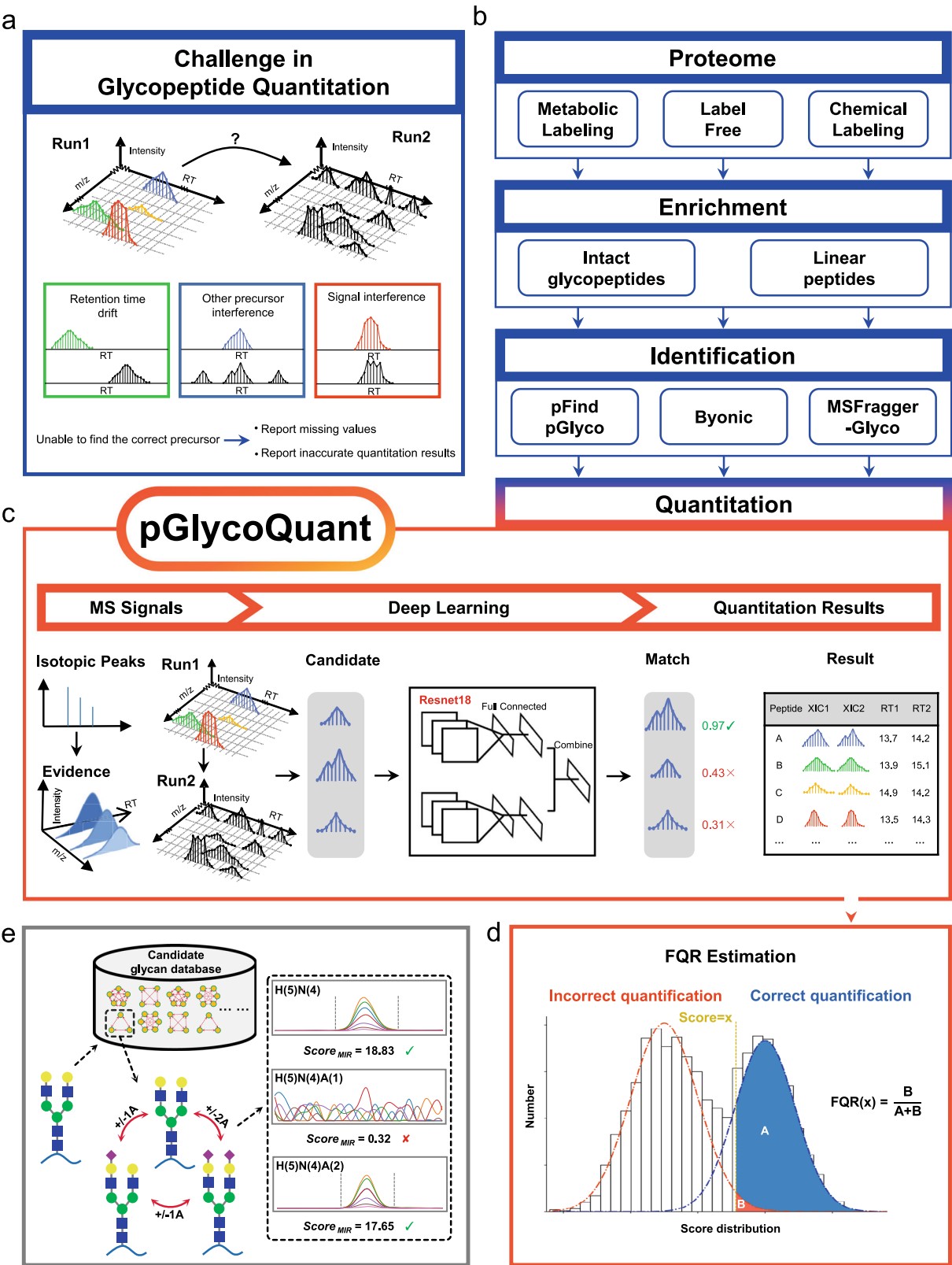

**Fig. 1 | The development of pGlycoQuant. a** Current glycopeptide quantitation software tools suffer from suboptimal reproducibility for lower-abundance signals, resulting in a high missing value rate. **b** pGlycoQuant supports both primary and tandem mass spectrometry quantitation for multiple quantitative strategies. **c** An embedded deep learning model of ResNet in pGlycoQuant for glycopeptide evidence matching and MBR analysis. **d** The FQR is estimated by fitting Gaussian mixture distribution with EM algorithm. **e** The MIR algorithm is proposed as an optional function in pGlycoQuant for increasing quantitative coverage of sialic acid-related glycopeptides.

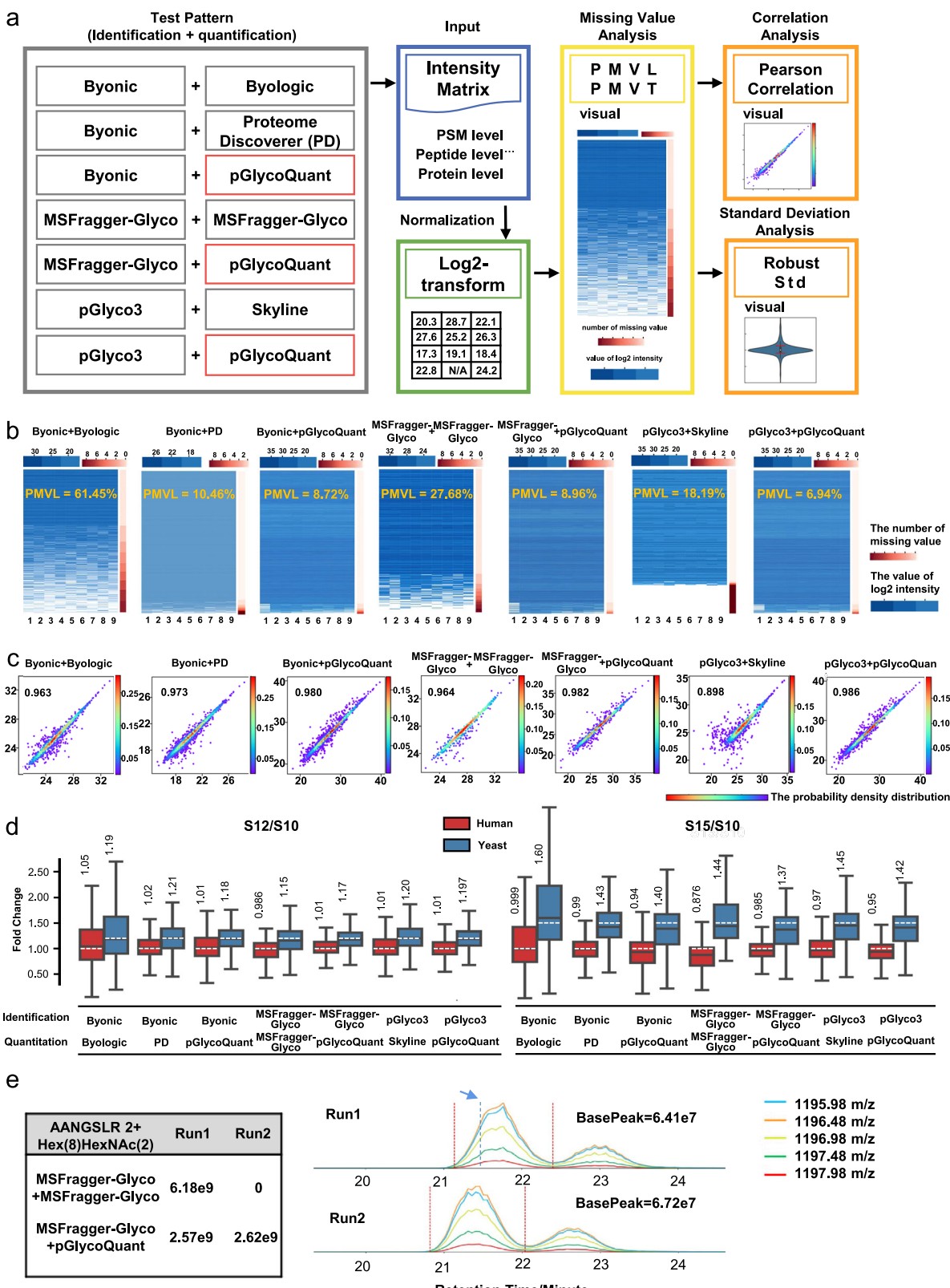

Table 4). By comparison, pGlycoQuant reduced the missing values by 19–86% in PMVL and 30–89% in PMVT, even when reading the same GPSMs reported by the other searching engines (Fig. 2b and Supplementary Table 4). The decline of missing values reported by pGlyco-Quant could be attributed to the efficient deep learning-based evidence matching method that efficiently extracts the glycopeptide signals (Supplementary Fig. 7), which could enable highly reproducible quantitation in large sample cohorts. For the SILAC-labeled data, pGlycoQuant also showed outstanding performance on reducing missing values compared with other tools (Supplementary Table 4). Compared to SILAC-labeled and label-free data, it is relatively easy to obtain quantitation results from TMT-labeled data, as only report ions should be considered. Unfortunately, Byologic currently does not support the quantitation of TMT-labeled data. Although Proteome

**Fig. 2 | Evaluation of pGlycoQuant performance on intact glycopeptide quantitation of label-free data. a** The comparison workflow. **b** pGlycoQuant reports fewer missing values than other software tool, even reading the same identification results. **c** Pearson correlation of two replicate runs from the same sample. **d** Box plot visualization of the fold change of the glycopeptide quantification results of the mixed-organism samples. Percent changes were calculated based on the mean quantities in three replicates of each sample, including results with partially missing values. The medians are indicated. The boxes indicate the interquartile ranges (IQRs), and the whiskers indicate 1.5 × IQR values; no outliers are shown. The white dotted lines indicate the theoretical fold changes of the organisms (1:1:1 (S10:S12:S15) for humans and 1:1.2:1.5 (S10:S12:S15) for yeast). **e** An example of a glycopeptide signal. MSFragger-Glyco reported missing values, but pGlycoQuant correctly identified the glycopeptide signal. The table on the left shows the intensity of the glycopeptide reported by MSFragger-Glyco and pGlycoQuant. The plot on the right side shows the extracted ion currents (XICs) of the glycopeptide in the mass spectrometry data, the blue dotted line represents the retention time to identify the corresponding glycopeptide by MSFragger-Glyco, and the red dotted line represents the start and end times for pGlycoQuant quantification of the glycopeptide. pGlycoQuant can exactly determine the start and the end points along the chromatogram profiles, and match the expected evidences. Source data are provided as a Source Data file.

Discoverer, MSFragger-Glyco, and Skyline could analyze the TMT-labeled data, MSFragger-Glyco produced mediocre results on the PMVL and PMVT (4.52% and 4.52% at glycopeptide level, respectively), which was even inferior in both Proteome Discoverer (31.99% PMVL and 31.99% PMVT) and Skyline (11.64% PMVL and 11.64% PMVT). In contrast, pGlycoQuant only reported 0.16% PMVL and 0.16% PMVT (Supplementary Table 4).

After removal of the missing values, the quantitation precision was evaluated. Here, we use Pearson correlation and standard deviation as the measurements of precision. For SILAC-labeled data quantitation, it was shown that the quantitative results reported by pGlycoQuant had higher correlation and lower standard deviation compared with other tools. Especially, the Pearson correlation of Byologic was only 0.665, while the standard deviation of it was about one and a half times higher than that of the pGlycoQuant counterparts (Supplementary Fig. 8 and Supplementary Table 4). That is caused by the outlier results of the low intensity glycopeptides. For the label-free and TMT-labeled data, the pGlycoQuant results reported the Pearson correlation higher than 0.974 and 0.990, and the standard deviation lower than 0.386 and 0.105, respectively (Fig. 2c and Supplementary Figs. 9 and 10). Other software tools except Skyline also achieved comparable Pearson correlation and standard deviation, however, due to the removal of missing values during the analysis.

Then, we used mixed-organism sample data[31] and the above fold change-(de)glycoproteome data to assess the precision of quantification on the basis of how well the known ratios could be recovered by the software tools. It was demonstrated that pGlycoQuant showed better quantification precision for both of the mixed-organism sample data (Fig. 2d) and the fold change-(de)glycoproteome data (Supplementary Figs. 11–13). By visualizing extracted ion currents (XICs) of the glycopeptide in two repeated runs of label-free data, we showed that the quantitative algorithm of pGlycoQuant can accurately locate the signal of the glycopeptide (Fig. 2e).

The above results demonstrated that pGlycoQuant with the deep-learning-based evidence matching model has outstanding performance in quantitative analysis of intact glycopeptides. Moreover, the quality control in pGlycoQuant effectively removes low-quality quantitative data, further ensuring quantitative accuracy and precision.

### A MIR algorithm as an optional function in pGlycoQuant for more quantitative coverage of glycopeptides

We also proposed a MIR algorithm as an optional function in pGlycoQuant for increasing quantitative coverage of sialic acid (SA)-related glycopeptides as much as possible at current stage (Fig. 3a). A candidate glycan database was constructed from the two of maximum human glycan structures[32] and the glycans in the database were grouped into several subnets (Supplementary Fig. 14). As shown in Fig. 3b, an identified glycopeptide attached with a glycan that can be matched in any one of the subnets is used for MIR analysis. Glycopeptides attached with the glycans from the subnet were treated as glycopeptide candidates. Then pGlycoQuant constructs glycopeptide evidences in full MS scans based on those candidates' m/z and preset retention time (RT) shifts. The RT shift values can be either generated in real-time or self-defined by users. The theoretical isotope distribution of a constructed evidence is used as the template to match with its experimental isotope peaks in each full MS scans. Each matching is scored as cosine similarity ($Sim_{iso}$). The experimental isotope peaks with the >0.9 are combined along with the retention time dimension to form the evidence and summed up to calculate the MIR matching score of the evidence. Finally, the matched evidences with intensity and MIR matching score information are reported along with the pGlycoQuant results. Detailed MIR methods are described in "Methods" section.

We conducted experiments using benchmarked N-glycopeptides attached with or without SAs to validate the accuracy and sensitivity of MIR (Supplementary Fig. 15). We first synthesized five N-glycopeptides attached with the glycan H(5)N(4) (0SA-GPs) and five *N*-glycopeptides attached with the glycan H(5)N(4)A(2) (2SA-GPs), mixed them with yeast glycopeptides (all high-mannose types) to mimic complex samples, and performed MIR analysis on each of the following mixtures: (1) mixture 1 containing 0SA-GPs and yeast glycopeptides, (2) mixture 2 containing 2SA-GPs and yeast glycopeptides, (3) a series of mixtures containing 0SA-GPs and 2SA-GPs with different ratios (0SA-GPs:2SA-GPS, 1:5, 1:2, 1:1, 2:1, 5:1) and yeast glycopeptides. As shown in Fig. 3c, a 100% coverage and a high-correlation of quantitation results were got for the subsistent 0SA-GPs in the mixture 1. Meanwhile, MIR matching scores more than 5 were reported for 0SA-GPs in contrast to the MIR matching score (<5) reported for 1SA/2SA-GPs in the mixture 1 (Fig. 3d). Similarly, the results consistent with expectations were also got for that in the mixture 2 (Fig. 3e, f). It was noted that although three abnormal MIR matching score (>5) were reported for 1SA-GPs (Fig. 3f) that were not incorporated in the mixture 2, the isotope distribution of the evidence showed high similarity with the theoretical isotope distribution (Supplementary Fig. 16), suggesting that these 1SA-GPs were highly likely to exist in the mixture 2 probably due to the cleavage of SA during LC-MS/MS analysis.

Furthermore, MIR analysis was performed on a series of mixtures containing 0SA-GPs and 2SA-GPs with different abundance as well as yeast glycopeptides as interferences. The fold changes of detected glycopeptides in each mixture sample were calculated and visualized in Fig. 3g, h. The fold change of the 0SA-GPs in each mixture (Fig. 3g) and the 2SA-GPs in each mixture (Fig. 3h) reported by MIR analysis were closer to the theoretical values and were highly consistent with the quantitation without MIR analysis.

Finally, we applied the MIR analysis to a biological sample, human IgG, of which glycosylation plays an extremely important role in immune function and is closely related to many pathological processes[33]. Combining all the 3 replicates, a total of 245 glycopeptides with 103 glycans were quantified using pGlycoQuant with MIR matching score more than 5 (Supplementary Data 1 and Supplementary Fig. 17). Compared with pGlycoQuant analysis without MIR, 17 more glycopeptides with 10 glycans

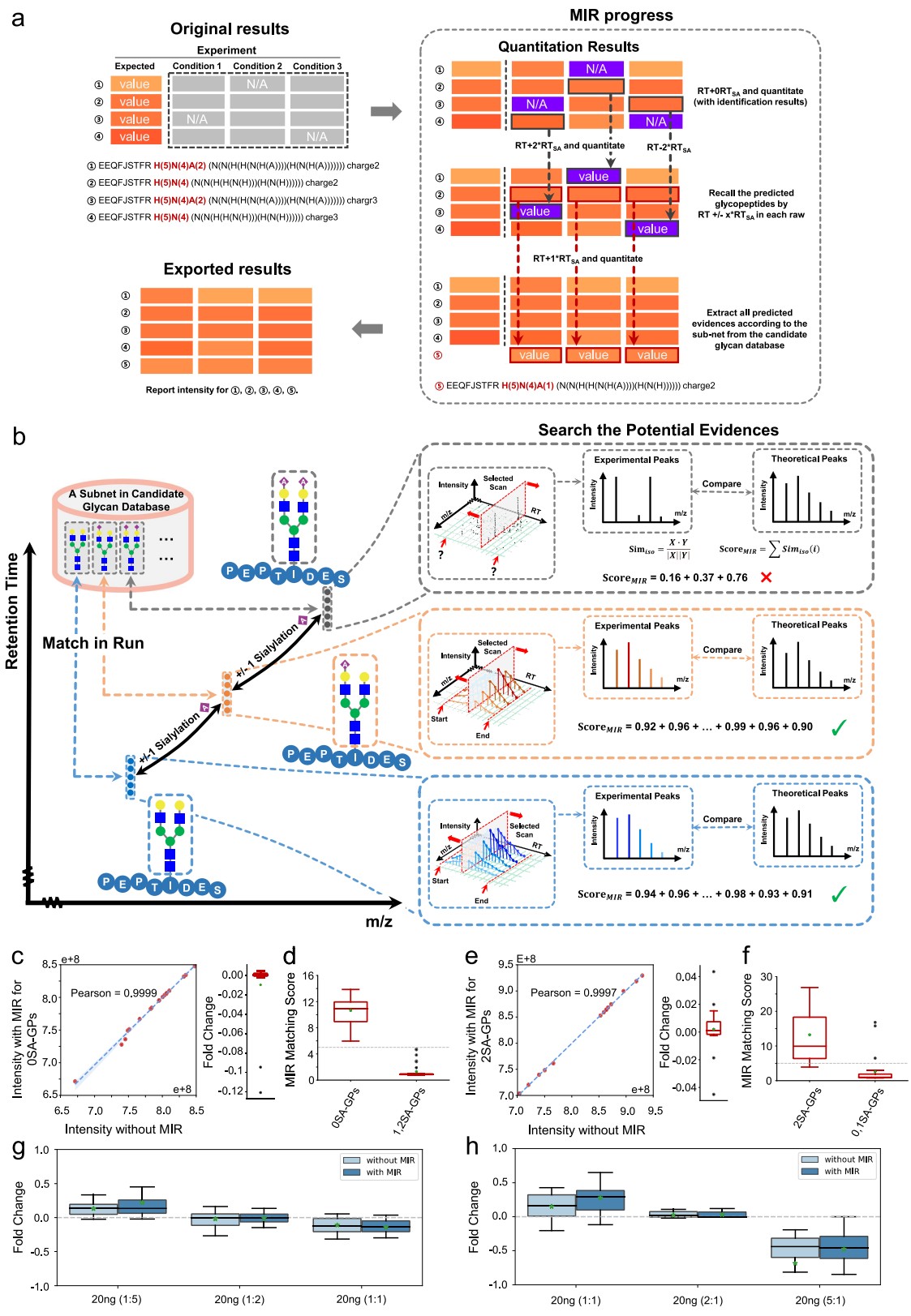

were quantified using MIR. Among them, 14 glycopeptides have been reported with tandem mass spectra evidences by other study[34], which conducted in large sample cohorts and employed an additional labeling followed by a combination of ETD and HCD analysis. The above analyses demonstrated the utility of the MIR in pGlycoquant for more quantitative coverage of glycopeptides.

## pGlycoQuant enabled large-scale quantitative analysis of the proteome and N-glycoproteome in different metastatic HCC cell lines

The high accuracy and precision of pGlycoQuant enable further functional exploration of site-specific glycosylation. Quantitative analyses of the proteome and intact N-glycopeptides in three HCC cell lines with different metastatic potentials (Hep3B with no metastatic

**Fig. 3 | A MIR algorithm proposed as an optional function in pGlycoQuant.**
**a** Overview of the MIR processing in pGlycoQuant. All glycopeptides are expected to be identified and quantified in one experiment, however, some of the glyco-peptides could not be observed in the practical experiments. MIR could achieve more quantitative coverage of glycopeptides within one experiment. **b** The work-flow of MIR. **c–h** Evaluation of the quantitative performance of MIR analysis on benchmarked N-glycopeptides. We used the data obtained from the experiments described in Supplementary Fig. 15 to validate the accuracy and sensitivity of MIR. The correlation of quantitative results reported by pGlycoQuant without MIR and with MIR for the subsistent standard glycopeptides, 0SA-GPs in mixture 1 (**c**) and 2SA-GPS in mixture 2 (**e**). MIR matching score reported by pGlycoQuant for mixture

1 (**d**) and mixture 2 (**f**). Box plot visualization of the fold change of the quantitation results of the 0SA-GPs (**g**) and 2SA-GPs (**h**) in a series of mixtures. Percent changes were calculated by dividing the intensity of each glycopeptide in each sample by the median intensity of this glycopeptide in all samples. The medians are indicated. The boxes indicate the interquartile ranges (IQRs), and the whiskers indicate 1.5 × IQR values; no outliers are shown. The gray dotted lines indicate the theore-tical fold changes of the benchmarked N-glycopeptides (1:1:1 for 0SA-GPs in the mixtures of 1:5, 1:2, 1:1 (0SA-GPs:2SA-GPS), and 1:1:1 for 2SA-GPs in the mixtures of 1:1, 2:1, 5:1 (0SA-GPs:2SA-GPS)). Each sample in the above experiment was analyzed by LC-MS/MS with 2 replicates (**c–f**) or 4 replicates (**g, h**). Source data are provided as a Source Data file.

potential, MHCC97L with low metastatic potential and MHCCLM3 with high metastatic potential) were performed with four replicates for each MS quantitative analysis (Fig. 4a). A total of 11,312 proteins and 11,001 intact N-glycopeptides were quantified (Supplementary Data 2 and 3), among which, those that appeared more than in duplicate were regarded as reliable. The results showed that a total of 9154 proteins and 6435 intact N-glycopeptides were reliably identified and quantified from the proteomic and intact N-glycopeptide quantitation experi-ments, respectively (Fig. 4b and Supplementary Data 2 and 3), which was the largest intact glycopeptide quantitative result in the three cell lines thus far. The 6435 intact N-glycopeptides were attributed to 769 glycoproteins with 1357 N-glycosites and 143 N-glycans (Fig. 4c). The quantitative ratios among the three cell lines showed high correlation, demonstrating reliable quantitative accuracy and good repeatability (Supplementary Fig. 18, Supplementary Fig. 19). The criteria of ratio ≥2 or ≤0.5 and $p < 0.01$ with multiple-testing correction were adopted as significant differential expression to further filter the quantitative results, resulting in 1429 proteins and 1940 intact glycopeptides. The details of the differential proteins and intact glycopeptides are listed in Supplementary Data 4 and Supplementary Data 5.

The ability to quantify the proteome and intact glycopeptides at such a large scale provides opportunities to investigate the role of glycosylation in the metastasis of HCC. Gene Ontology (GO) analyses of the proteome and glycoproteome showed that differential proteins were mainly concentrated in the cytoplasm and nucleus (Supplemen-tary Fig. 20a), were associated with ion binding and RNA/DNA binding (Supplementary Fig. 20b), and participated in cellular metabolism and signal transduction (Supplementary Fig. 20c), while differential intact N-glycopeptide-related glycoproteins were more likely to be located in the membrane and extracellular regions (Supplementary Fig. 21a), to be related to hydrolase activity (Supplementary Fig. 21b), and to be involved in cell adhesion (Supplementary Fig. 21c).

Then, we used volcano plots and box plots to visually show the distribution and dispersion degree of the differential proteins and intact glycopeptides. The differences in glycopeptides were more diffuse than those in proteins (Fig. 4d–g) in the three cell lines. Further principal component analysis (PCA) showed that compared to pro-teomes, differences in intact glycopeptides were more likely to dis-tinguish the three HCC cell lines with different metastatic potentials (Fig. 4h, i). Thus, we drilled down for in-depth information on intact glycopeptide data to explore the role of site-specific N-glycosylation in the metastasis of HCC.

### Site-specific N-glycoproteomic analyses revealed great hetero-geneity and implied altered core fucosylation to be highly associated with in vitro cell invasion and metastasis
Statistical analyses of the site-specific N-glycoproteome enable further visualization of glycoproteome heterogeneity and investigation of system-wide glycosylation patterns[35]. Firstly, we performed overall statistical analyses on site-specific N-glycoproteome data from three cell lines. It was demonstrated that ~79.5% of the glycosites (1079 of the 1357) quantified in this study were annotated in the UniProt database

(Fig. 5a). In addition to quantifying 456 previously proven glycosties (456 published and 48 imported), we provided experimental evidence for 575 UniProt-predicted glycosites (572 sequence analyses and 3 by similarity) and 278 non-UniProt-recorded glycosites (Fig. 5a). Sequence motif analysis showed that the majority of N-glycosites share N-X-S (40%) and N-X-T (58%) sequons, while only 2% of the glycosites have the N-X-C sequon (Fig. 5b). The distribution of singly or multiply glycosylated proteins and the degree of glycan microheterogeneity showed that more than half of the glycoproteins (481 of the 769 identified glycoproteins) had only one glycosite (Fig. 5c), while 75% of glycosites contained more than one glycan (Fig. 5d). Glycans with 8–12 monosaccharides dominated in these data (Fig. 5e). A network between glycan types and glycosites on glycoproteins revealed that the fucosylation type was prevalent in the HCC cells, and fucosylation and sialylation occurred more frequently on multiply-glycosylated proteins, thus contributing more to heterogeneity (Fig. 5f). A heatmap displaying the frequency of glycan pairs co-occurring at the same site illustrated that oligomannose appears to co-occur with several groups of complex/hybrid, fucosylation and sialylation types with high frequency (Fig. 5g), which further indicates site-specific microheterogeneity.

We further analyzed different distributions of glycan size and glycan type in all quantified intact glycopeptides and the uniformly up-/downregulated glycopeptides in three cell lines with increased metastatic potential. It could be concluded that upregulated glyco-peptides tend to have longer glycans (mostly with 8–12 mono-saccharides) than downregulated glycopeptides (Supplementary Fig. 22a). Comparing glycan types in all quantified glycopeptides showed that fucosylation and sialylation were more associated with upregulated glycopeptides, while oligomannose was dominant in downregulated glycopeptides (Supplementary Fig. 22b).

Glycosyltransferases (GTs) and glycoside hydrolases (GHs) cor-egulate the synthesis of glycans and are key factors affecting protein glycosylation. Thus, we then analyzed glycan-related enzymes. We quantify 161 glycan-related genes, including 84 GTs and 43 GHs, from the proteome quantitative results of 9154 proteins (Supplementary Fig. 23 and Supplementary Data 6). Among them, we noted that a GT that regulates core fucosylation synthesis, alpha-(1,6)-fucosyltransfer-ase (FUT8), was significantly changed in the three cell lines (Supple-mentary Fig. 23 and Supplementary Data 6), which implies that core fucosylation is highly correlated with HCC cell metastasis.

### Site-979-specific core fucosylation of L1CAM was identified and validated in vitro as a potential regulator of HCC metastasis
Consequently, we further analyzed core-fucosylated glycoproteins, and our screen identified a glycoprotein, L1CAM, in which glycosite 979 is highly core-fucosylated and upregulated in three cell lines with increasing metastatic potential. A total of 35 site-specific glycans, including 5 glycosites and 20 glycans were quantified in L1CAM (Fig. 6a, b). L1CAM is a highly glycosylated protein known to reg-ulate cell attachment, invasion and migration in several cancers and is associated with poor prognosis[36,37]. For example, Mahal and

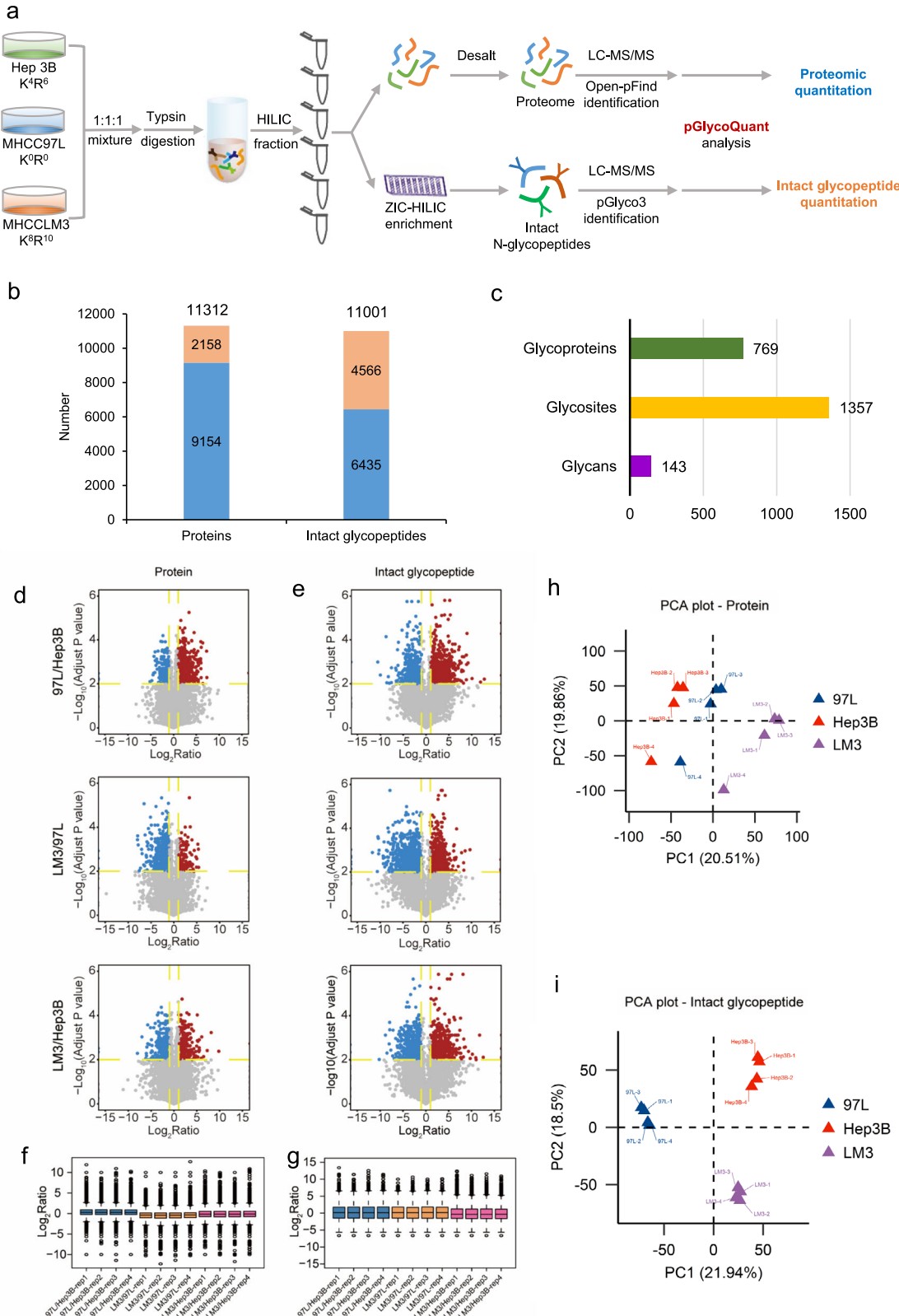

Hernando et al.[38] demonstrated that glycoprotein targets of FUT8 were enriched in cell migration proteins, including the adhesion molecule L1CAM, in melanoma metastases. However, little is known about the site-specific glycosylation of L1CAM associated with cell invasion and metastasis. We found that core fucosylation with glycan Hex[5]HexNAc[4]NeuAc[1]Fuc[1] at glycosite 979 was significantly high in L1CAM in all three cell lines (Supplementary Fig. 24a) through normalization within one cell line. After normalization among the three cell lines, it was obvious that all fucosylated glycans at site 979 of L1CAM were consistently upregulated with increasing metastatic potential of the cell lines (Supplementary Fig. 24b). Further analysis of the protein expression levels of L1CAM and FUT8 from proteomic quantitation data revealed that the upregulated of fucosylation at site 979 of L1CAM with increasing metastatic potential was caused by different reasons

**Fig. 4 | Large-scale quantitative analyses of the proteome and *N*-glycoproteome of different metastatic HCC cell lines with pGlycoQuant. a** Experimental workflow. **b** The numbers of proteins and intact glycopeptides quantified from proteomic and intact *N*-glycopeptide quantitation experiments, respectively. The bar in blue represents the number of proteins/glycopeptides quantified in more than two replicates, and the bar in orange represents the number of proteins/glycopeptides quantified in <3 replicates. **c** General information on 6435 reliably quantified intact *N*-glycopeptides. Volcano map of differential proteins (**d**) and intact glycopeptides (**e**) among three cell lines. Significant differential expression in

the volcano plots were calculated by two-sided *t*-test and the false discovery rate (FDR)-adjusted P-values were calculated by Benjamini–Hochberg (BH) FDR-controlling method for multiple testing. The detailed data are shown in the Supplementary Data 4 and 5. Box diagram with four replicates for the quantitation of proteins (**f**) and intact glycopeptides (**g**). In the box plots of f and g, the center line indicates the median. The boxes indicate the interquartile ranges (IQRs), and the whiskers indicate 1.5 × IQR values (with outliers shown). Principal component analysis (PCA) plot of differential proteins (**h**) and differential intact glycopeptides (**i**) among three cell lines.

(Fig. 6c): from no metastatic potential to low metastatic potential, the increased fucosylation at site 979 was caused by the increased expression of FUT8; from low metastatic potential to high metastatic potential, the increased fucosylation at site 979 was mainly due to the increased protein content of L1CAM. A comparison of differential intact glycopeptides without and with normalization to protein abundance also showed that increased fucosylation abundance at site 979 was caused by the protein abundance changes from low metastatic potential to high metastatic potential (Supplementary Fig. 25 and Supplementary Data 7). The western blot results were consistent with the of MS-based proteomic quantitative results interpreted by pGlycoQuant (Fig. 6d–f). Based on the above results and the known ability of L1CAM support invasion and metastasis, we hypothesize that increased core fucosylation at site 979 of L1CAM reduces L1CAM cleavage by plasmin, facilitating HCC cell line invasion and metastasis (Fig. 6g).

We performed several experiments to investigate the impact of site-979-specific core fucosylation of L1CAM on in vitro HCC cell metastasis. We first examined whether L1CAM is required for the maintenance of existing metastasis by the silencing of L1CAM in LM3 cells. Consistently, siL1CAM cells displayed decreased L1CAM protein (Supplementary Fig. 26a) and reduced cell migration and invasion in comparison to siCtrl cells (Supplementary Fig. 26b–d). To confirm the role of site-979-specific core fucosylation of L1CAM in vitro HCC cell metastasis, we next investigated whether L1CAM overexpression with or without the site-979 mutation has the same ability to promote HCC cell metastatic capacity and whether the core fucosylation of L1CAM contributes to that effect. L1CAM overexpression triggered significant increases in 97 L cell migration and invasion in vitro, while site-979-mutated L1CAM overexpression with the same protein amount showed no prometastatic effects (Fig. 7a–d). Further silencing of FUT8 in 97L cells (Fig. 7e), which resulted in reduced core-fucosylated L1CAM (Fig. 7f), decreased in vitro cell migration and invasion (Fig. 7g–i). These results suggest that site-979-specific core fucosylation is critical to prometastatic phenotype in HCC cell lines. Previous studies have reported that the cleavage of L1CAM by plasmin inhibits its ability to mediate neural cell invasion and metastatic outgrowth[39,40]. Here, we observed that 97L cells with site-979-mutated L1CAM overexpression indeed tended to be more easily cleaved by plasmin than unmutated cells (Fig. 7j), which to a certain extent accounted for the impact of altered core fucosylation at site 979 of L1CAM on L1CAM cleavage by plasmin.

## Discussion

Since the identification of intact glycopeptide has been greatly facilitated by software tools[6,41,42], there is an urgent need to develop efficient tools for accurate intact glycopeptide quantitation to assist in exploring differences in site-specific glycosylation[7,20]. The main challenge in accurate quantitation by LC-MS/MS-based methods is to correctly extract the targeted mass spectral signal since it is easily interfered with nearby signals or noise. Herein, we developed pGlycoQuant to support multiple common glycopeptide quantitative strategies. pGlycoQuant applies a ResNet deep learning model to process glycopeptide quantitative evidence between or within MS

runs. The superiority of ResNet in solving the image recognition makes it great potential in recognition of chromatographic curves and could also be used for proteomic quantification to solve missing value problem, but only if the dedicated model is trained. In this work, the ResNet model was used to learn the in-depth representation of glycopeptide quantitative evidence in complex mass spectrometry data, improving the sensitivity and precision in detecting low-abundance glycopeptides signals. Moreover, the deep-learning model reporting the matching score, which is difficult to measure by the traditional algorithms, is used to calculate of false quantitation rate for quality control of the quantitation results. We benchmarked our pGlycoQuant with several prevalently used software tools on different quantitation strategy-based datasets, and the results demonstrated that pGlycoQuant outperforms other tools (Supplementary Table 2) in terms of precision and reproducibility.

The use of MBR in pGlycoQuant could improve the problem of random precursor selection for MS2 analysis in different replicates, reduces missing values and increases quantitative reproducibility. In addition to the MBR algorithm, we also proposed a MIR algorithm as an optional function in pGlycoQuant to get around the issue of inadequate precursor selection for MS2 analysis in data dependent acquisition (DDA)-based intact glycopeptide analytical strategies in some extent. At current stage, pGlycoQuant MIR could only be used for more quantitative coverage of sialic acid-related glycopeptides. The evaluation of the MIR using benchmarked *N*-glycopeptides and the application in human IgG showed fine accuracy and sensitivity of pGlycoQuant with MIR analysis. While, it is noted that MIR is untested on truly complex mixtures and should be thoroughly evaluated manually any time it is used. The MIR algorithm proposed here, though has a lot of room for improvement, shows a potential ability of quantitation covering more results than identification.

pGlycoQuant for glycoproteome quantitation at the site-specific glycosylation level provides us with opportunities and horizons to explore the role of glycosylation organisms. The combination of large-scale quantitative analyses of the proteome and glycoproteome in three different metastatic HCC cell lines demonstrates a generic application of pGlycoQuant for investigating the role of site-specific glycosylation, yielding the largest intact glycopeptide quantitative data in three HCC cell lines and enabling the visualization of glycoproteome heterogeneity and the investigation of system-wide glycosylation patterns. Based on the convincing quantitative results obtained by pGlycoQuant, fortunately, the site-979-specific core fucosylation of L1CAM was identified in a screen and validated as a potential regulator of HCC metastasis in vitro, which presents the possibility of pGlycoQuant in biological research.

Currently, pGlycoQuant is compatible with many search engines, including Open-pFind, pGlyco2.0, pGlyco3, MSFragger-Glyco, and Byonic, for glycoproteome quantitation at intact glycopeptide level. Although pGlycoQuant is shown here in the context of *N*-glycoproteomic quantitation, it is also applicable to intact O-glycopeptide quantitation. With a deep residual network reducing missing values by over 60% compared with other quantitative software tools, pGlycoQuant makes it possible to quantitatively investigate site-specific glycosylation and illuminate its functions.

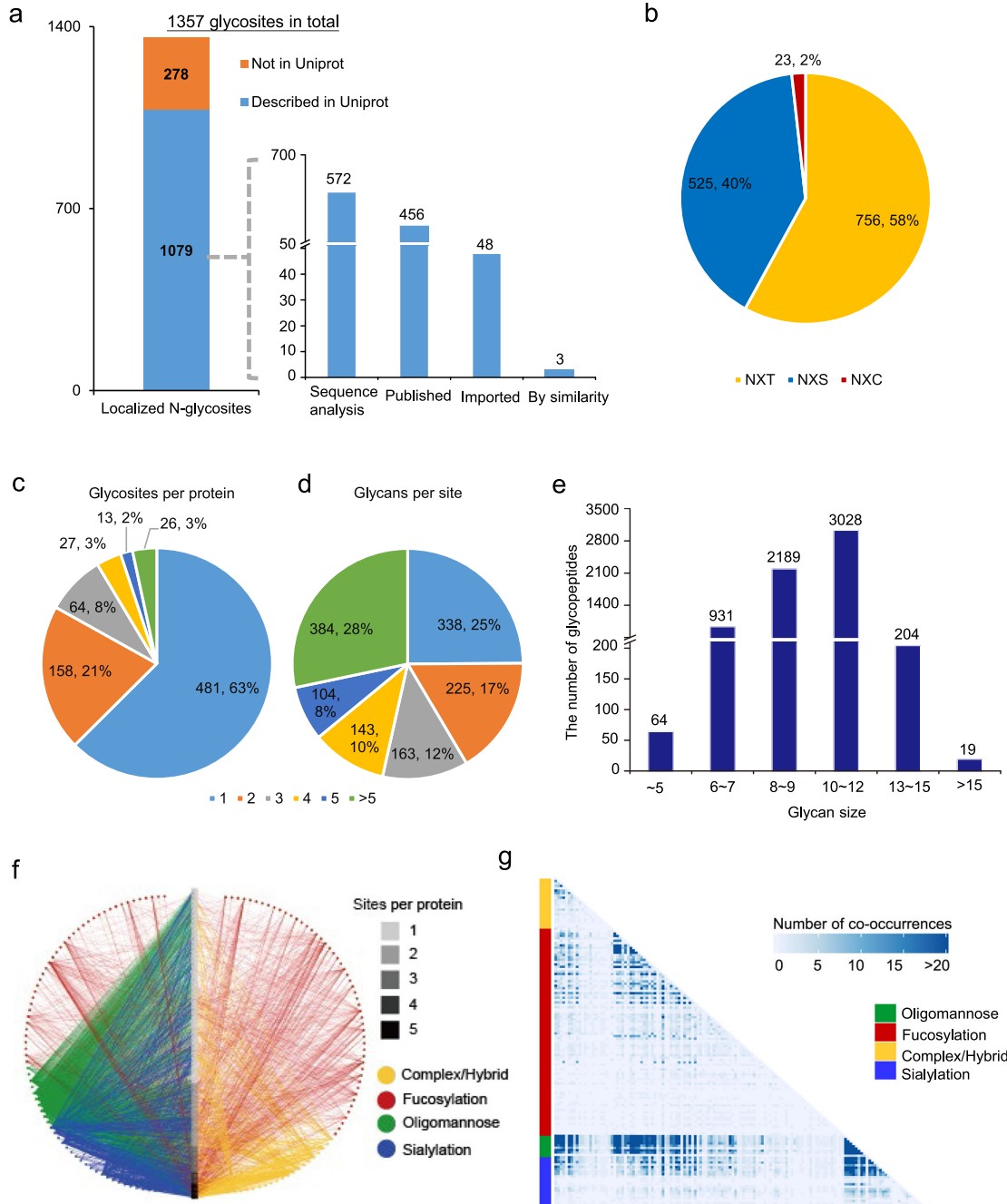

**Fig. 5 | Characteristics of site-specific *N*-glycans quantified in HCC cell lines.**
**a** The status of quantified glycosites recorded in the UniProt database.
**b** Recognition of the sequence motif of *N*-glycosylation (N-X-S/T/C, where N is asparagine, X is any amino acid except proline, S is serine, T is threonine, and C is cysteine). **c** The number and percentage of *N*-glycosites located on a certain protein. **d** The number and percentage of *N*-glycans linker to a certain glycosite. **e** The glycan size distribution in 6,435 glycopeptides from three HCC cell lines.
**f** Glycoprotein-glycan network maps of specific glycans (outer circle, 143 in total) modifying specific glycoproteins (inner bar, 769 in total). Glycoproteins are sorted

by number of glycosites (scale to the right). Glycans are organized by classification. **g** A glycan co-occurrence heatmap representing the number of times glycan pairs appear together at the same glycosite, indicating which glycans contribute most to the microheterogeneity of the 1019 glycosites with more than one glycan modifying them. In the glycan classification, Complex/Hybrid is for the complex/hybrid-type glycans with no fucose and sialic acid. Oligomannose is for the glycans only containing mannose monosaccharide except for the core structure. Fucosylated glycans that contain sialic acid moieties are classified as Fucosylation instead of Sialylation.

## Methods
### Workflow of pGlycoQuant
As shown in Supplementary Fig. 1, the workflow of pGlycoQuant consists of three steps:

Step 1: Reading the identification results. pGlycoQuant can read the identification results from pGlyco, Byonic, and MSFragger-Glyco.

High confidence GPSMs produced by identification software tools can be read into the program.

Step 2: Extracting signals. pGlycoQuant constructs chromatograms for individual isotopic peaks of the glycopeptides. These "isotopic chromatograms" are named as "evidence" in this paper. For each input GPSM, pGlycoQuant calculates the theoretical distribution of

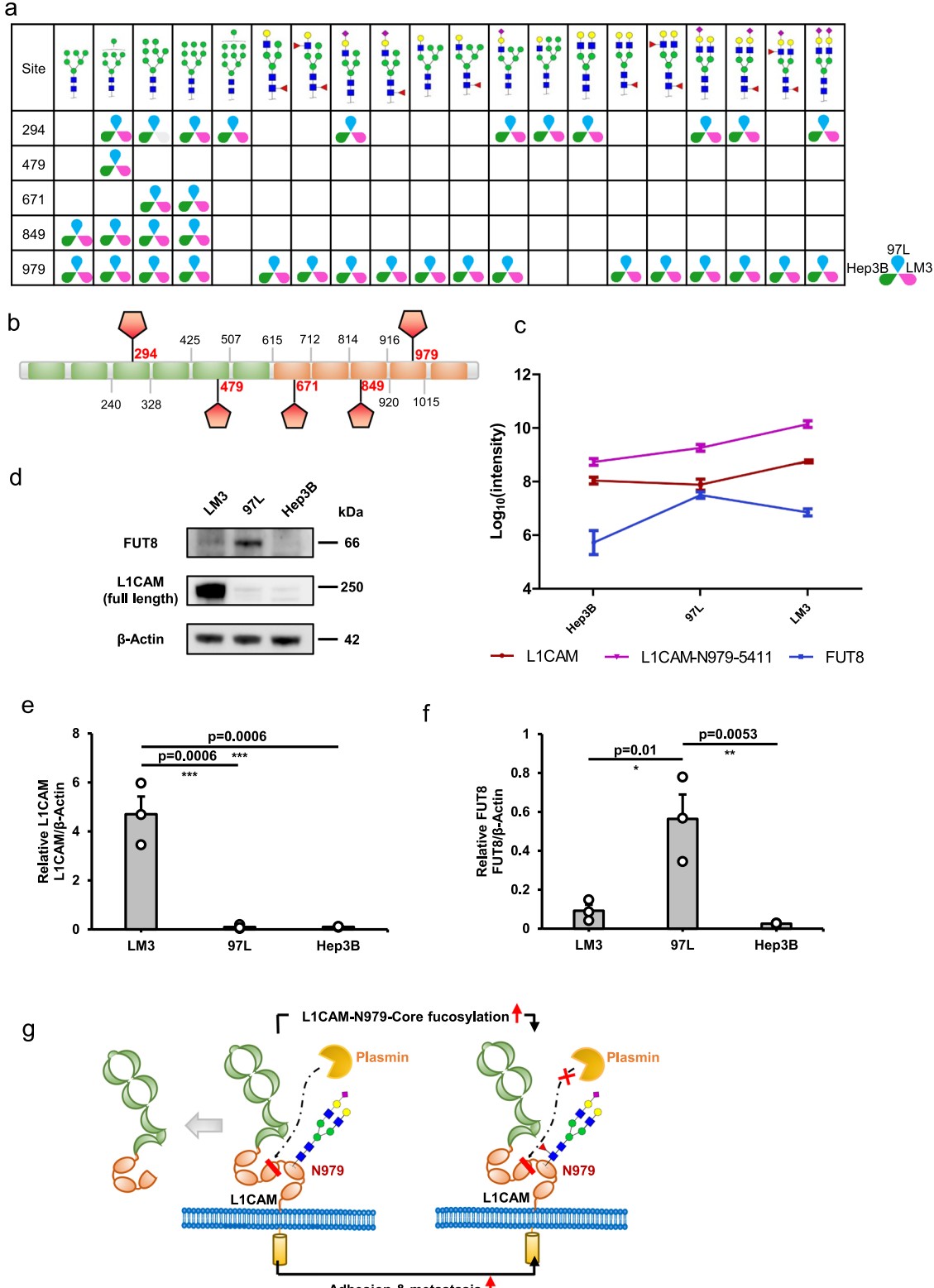

**Fig. 6 | Site-979-specific core fucosylation of L1CAM is upregulated in three HCC cell lines with increasing metastatic potential. a** Site-specific *N*-glycosylation of L1CAM in three HCC cell lines. **b** Graphical view of five identified glycosites located in specific positions and domains of L1CAM. The orange bar chart represents the Ig-like C2 type, and the green bar chart represents fibronectin type III. **c** Change trends of protein L1CAM, site-979-specific glycan Hex[5]HexNAc[4]NeuAc[1]Fuc[1] (L1CAM-N979-5411), and protein FUT8 in three HCC cell lines from no metastatic potential to high metastatic potential. Data are presented as mean values ± SD. (*n* = 4 biological replicates) (**d**) Western blot validation of the protein expression levels of L1CAM and FUT8 in three HCC cell lines. **e** L1CAM levels in three HCC cell lines. (f) FUT8 levels in three HCC cell lines. The data (**e**, **f**) are presented as mean values ± SEM from three independent experiments. P values were determined by one-way ANOVA with Tukey's test. **p* < 0.05, ***p* < 0.01, and ****p* < 0.001 compared to the control (*n* = 3). **g** Hypothesis of increased core fucosylation at site 979 of L1CAM facilitating HCC cell line invasion and metastasis. The glycan symbols are as follows: green circle for Hex, blue square for HexNAc, purple diamond for sialic acids and red triangle for fucose. See the Source Data file for the exact *P* values. Source data are provided as a Source Data file.

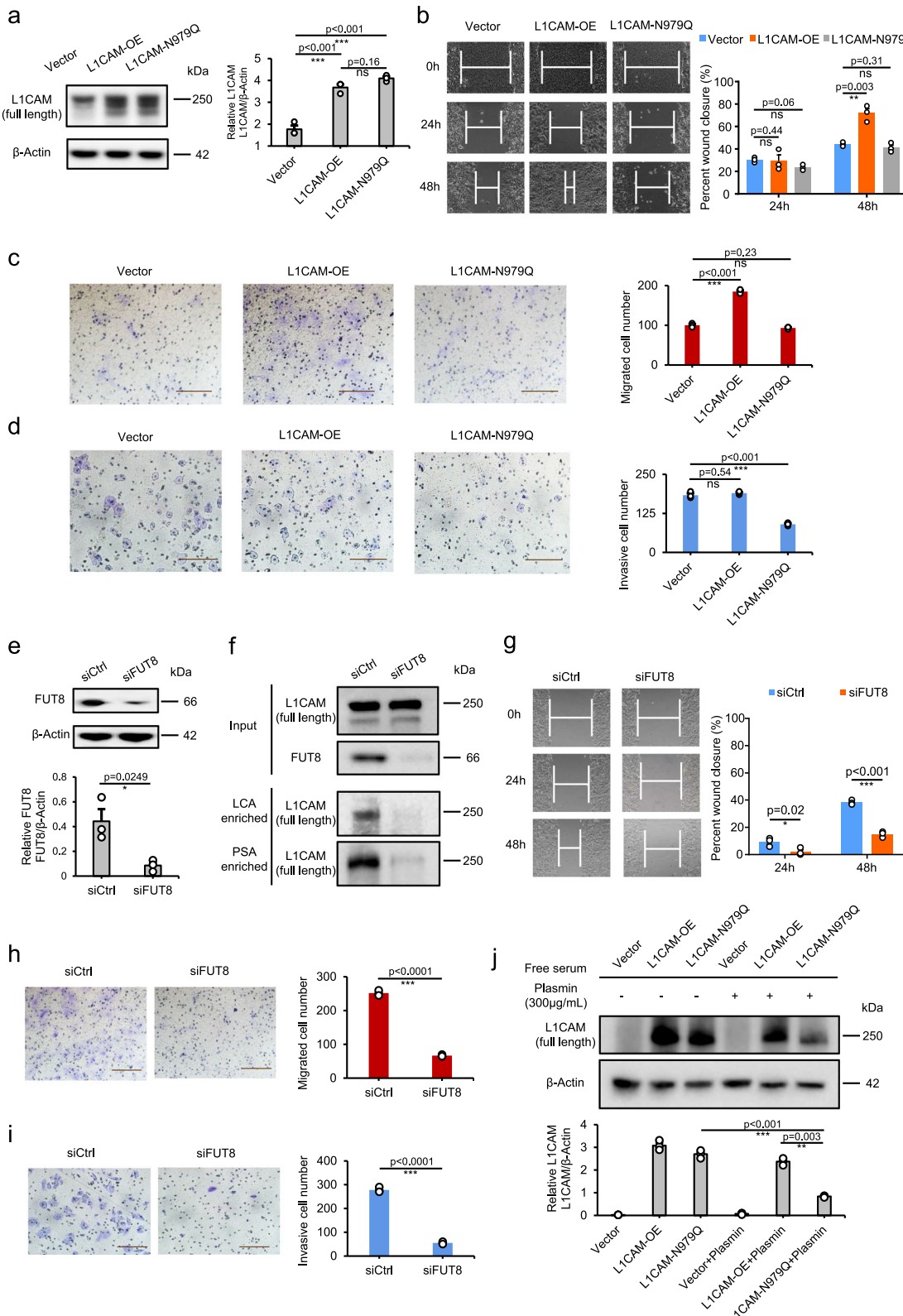

isotopic peaks using a stepwise convolution algorithm emass[43] and identifies experimental isotopic peaks in a range of full MS scans where the glycopeptide may be expected. For each MS scan, pGlycoQuant applies a ppm-level m/z tolerance window (normally ±10–±20 ppm, can be defined by users) around the theoretical m/z values of the isotopic peaks of the glycopeptide to select the experimental peaks. These experimental intensities along the retention time axis in contiguous MS scans are assembled into a chromatogram. This chromatogram extends both left and right from the trigger MS scan until the intensity drops below 10% of the apex of the extending profile. For the chemical labeling data, pGlycoQuant picks the reported ion peaks in MS2 scans according to the input parameters.

Step 3: Quantitation processing. pGlycoQuant runs different processes for the data obtained by different quantitative strategies. (1)

**Fig. 7 | In vitro validation of site-979-specific core fucosylation of L1CAM is a potential regulator of HCC metastasis. a** Western blot of L1CAM levels in 97 L cells overexpressing control vector, pL1CAM-FLAG vector or pL1CAM (N979Q)-FLAG vector. **b** Wound healing assay in monolayers of 97L-overexpressing cells (vector, L1CAM-OE and L1CAM-N979Q). The scratch area of the cells was detected with an inverted microscope (10×). **c** Transwell migration assay and **d** Matrigel invasion assay (scale bar = 100 μm) in 97L- overexpressing cells (vector, L1CAM-OE and L1CAM-N979Q). **e** Western blot of L1CAM levels in 97 L cells transfected with negative control (siCtrl) and FUT8 siRNA (siFUT8). **f** Lectin enrichment in the lysate of 97 L cells transfected with Ctrl or FUT8 siRNA followed by western blot with L1CAM antibody. Input showed no effect of FUT8 knockdown on L1CAM expression. PSA and LCA in the figure stand for the agarose bound Pisum Sativum Agglutinin lectin and agarose bound Lens Culinaris Agglutinin lectin that were used for specific enrichment of core-fucosylated glycoproteins. **g** Wound healing assay

in monolayers of 97L-knockdown cells (siCtrl and siFUT8). The scratch area of the cells was detected with an inverted microscope (10×). **h** Transwell migration assay and **i** Matrigel invasion assay (scale bar = 100 μm) in 97L-knockdown cells (siCtrl and siFUT8). **j** Western blot of L1CAM in whole lysates of 97L-overexpressing cells (vector, L1CAM-OE and L1CAM-N979Q) after treatment with plasmin. Cells were transfected for 24 h and then incubated with plasmin (300 μg/mL) for 8 h. The grayscale values of the western blot data (**a, e, j**) and scratch area of the wound healing assay were measured by Image J. β-Actin was used for the normalization of loading in all western blot data (**a, e, j**). The data are presented as mean values ± SEM from three independent experiments. *P* values were determined by one-way ANOVA with Tukey's test or the two-tailed unpaired *t*-test. ns not significant, *$p < 0.05$, **$p < 0.01$, and ***$p < 0.001$ compared to the control ($n = 3$). See the Source Data file for the exact *P* values. Source data are provided as a Source Data file.

Metabolic-labeling data. pGlycoQuant calculates the similarity based on the well-trained deep-learning-based evidence matching model between the light and heavy glycopeptides. The chromatogram area of light and heavy glycopeptides is recorded as the glycopeptide intensity. (2) Chemical-labeling data. A glycopeptide intensity is calculated by summing the intensities of the reported ion peaks of the corresponding GPSMs. (3) Label-free data. Two different algorithms were embedded in pGlycoQuant for quantitative processing of label-free data, Match between Run (MBR) and Match in Run (MIR). In MBR, for each identified glycopeptide in one run, pGlycoQuant detects the corresponding evidence in other runs, i.e., matches evidence between runs. Given an identified glycopeptide in one run, pGlycoQuant constructs its evidence (termed reference evidence) in the full MS scans and then calculates the matching scores of all the evidences (termed candidate evidence) with the same precursor mass in a ± 2 min retention time window (default, or can be defined by users) in another run. From the start to the end of the retention window, pGlycoQuant enumerates isotopic chromatograms as the candidate evidence, and calculates the matching scores between reference evidence and each candidate evidence. This matching score of two pieces of evidence is calculated based on the same well-trained deep-learning-based evidence matching model and the pair of pieces of evidence with the maximum matching score is selected to obtain the quantitation result between different runs. Finally, the quantitative quality control was automatically performed to estimate the false quantitation rate (FQR) of the quantitation results. In MIR, pGlycoQuant constructs glycopeptide evidences in full MS scans for the glycopeptide candidates that were produced from the identified glycopeptides within the preset retention time window through comparison the isotope distribution of the theoretical one with the experimental one within a run.

## A deep-learning-based evidence matching model

As illustrated in Supplementary Fig. 2, we used label-free data to train the deep learning model. The selection metric for the training set are as follows: If a glycopeptide is identified in two runs with a retention time difference <2 min and an intensity difference <±30%, the glycopeptide evidences of the glycopeptide from the two runs are selected as a positive pair. If a glycopeptide is identified in one run and meanwhile not identified in the other run, the glycopeptide evidence in the identified run and the random evidence in the unidentified run with a retention time difference of 4 min and different precursor masses are taken as a negative pair. A total of 3000 positive pairs and 3000 negative pairs were randomly selected from the label-free data as training set and used to train the following deep learning model.

We consider the evidence as a matrix, similar to a picture in computational vision. The matrix is then transformed to a $512 \times 1$ vector by the ResNet18 model. This transformation is the critical operation to measure the pattern of a glycopeptide evidence. The two vectors from two pieces of evidence are then combined into a $1024 \times 1$ vector, which is the input of a fully connected neural network. This

network is designed to describe the similarity of the two original pieces of evidence, and a $16 \times 1$ vector is its output. Moreover, given a pair of pieces of evidence, 10 classical features are also extracted as a $10 \times 1$ vector. Another fully connected neural network with a softmax loss function is designed to output the final matching score of the two original pieces of evidence. This final matching score is in the interval of [0, 1], where 0 corresponds to very dissimilar and 1 to very similar evidence.

## FQR calculation

The specific process of FQR calculation is as follows: (1) Quantitation of target and decoy glycopeptide spectra. The decoys were generated in silico from the target GPSMs list by adding a mass shift of 10 Da to the mass of each precursor. Then, glycopeptide spectra from both target and decoy were quantitatively analyzed by pGlycoQuant to obtain the matching score of each spectrum. (2) The use of EM algorithm to fit the Gaussian mixture distribution. The Gaussian distribution density function of true-positive set, $f_1(x)$ and false-positive set, $f_0(x)$, as well as the corresponding mixture probability $\pi_1$ and $\pi_0$, were fitted by the EM algorithm.

Four steps as follows are needed to train a Gaussian mixture model (the pseudo-code as below in step 4):

Step 1 Initialization of Gaussian mixture model (GMM)

The calculation formula of Gaussian mixture model is as follows:

$$f(x) = \pi_0 f_0(x) + \pi_1 f_1(x) = \pi_0 \frac{1}{\sqrt{2\pi\sigma_0^2}} \exp^{-\frac{1}{2\sigma_0^2}(x-\mu_0)^2} \\ + \pi_1 \frac{1}{\sqrt{2\pi\sigma_1^2}} \exp^{-\frac{1}{2\sigma_1^2}(x-\mu_1)^2} \tag{1}$$

Where $\pi_0$ and $\pi_1$ represent the mixed probability of false-positive set and true-positive set respectively, $\mu_0$, $\mu_1$ and $\sigma_0$, $\sigma_1$ represent the mean value and the standard deviation of them two.

For generating a better GMM, k-means algorithm is used to cluster the two parts according to the matching score, and calculate the parameters to fill in this model, such as the mixed probability $\pi$, the mean value $\mu$ and the standard deviation $\sigma$.

Step 2 Expectation of EM algorithm

Based on the currently estimated parameters, calculate the probability of being in the false-positive set or being in the true-positive set for each quantitation result.

For one score $x_i$, Bayes' theorem is used to calculated the probability $p$ of being the part $T_i$.

$$p_{0i} = P(T_i = 0 | X_i = x_i) = \frac{\pi_0 f_0(x_i)}{\pi_0 f_0(x_i) + \pi_1 f_1(x_i)} \tag{2}$$

$$p_{1i} = P(T_i = 1 | X_i = x_i) = \frac{\pi_1 f_1(x_i)}{\pi_0 f_0(x_i) + \pi_1 f_1(x_i)} \tag{3}$$

Step 3 Maximization of EM algorithm

Update the parameters $(\hat{\pi}_0, \hat{\mu}_0, \hat{\sigma}_0, \hat{\pi}_1, \hat{\mu}_1, \hat{\sigma}_1)$ calculated in step2 in the GMM.

$$\hat{\pi}_k = \frac{\sum_{i=1}^{N} p_{ki}}{N}, k \in \{0,1\} \tag{4}$$

$$\hat{\mu}_k = \frac{\sum_{i=1}^{N} p_{ki} x_i}{\sum_{i=1}^{N} p_{ki}}, k \in \{0,1\} \tag{5}$$

$$\hat{\sigma}_k = \frac{\sum_{i=1}^{N} p_{ki} (x_i - \hat{\mu}_k)^2}{N \sum_{i=1}^{N} p_{ki}}, k \in \{0,1\} \tag{6}$$

Step 4 Iteration of EM algorithm

Repeat steps 2 and 3 until the specified number of cycles (500 by default) or the parameters converge. End the EM algorithm.

**EM-Algorithm in pGlycoQuant.**

Input $\vec{x}$ (Matching scoring set of target and decoy library output by ResNet model)

$i \leftarrow 1$ *and convergence* $\leftarrow$ *False* (Cycle one)
$\hat{\pi}_{0,i}, \hat{\pi}_{1,i}, \hat{\mu}_{0,i}, \hat{\mu}_{1,i}, \hat{\sigma}_{0,i}, \hat{\sigma}_{1,i} \leftarrow \text{InitializeByK-means}(\vec{x})$

**while** $i < 500$ *and convergence* == *False* **do**
 $i \leftarrow i + 1$
 **E-step:**
 $p \leftarrow \text{EstimateLikelihood}(\hat{\pi}_{0,i-1}, \hat{\pi}_{1,i-1}, \hat{\mu}_{0,i-1}, \hat{\mu}_{1,i-1}, \hat{\sigma}_{0,i-1},$
 $\hat{\sigma}_{1,i-1}, \vec{x})$
 **M-step:**
 $\hat{\pi}_{0,i}, \hat{\pi}_{1,i}, \hat{\mu}_{0,i}, \hat{\mu}_{1,i}, \hat{\sigma}_{0,i}, \hat{\sigma}_{1,i} \leftarrow \text{Estimate Parameter}(p, \vec{x})$
 **if** $|\hat{\pi}_{0,i}, \hat{\pi}_{1,i}, \hat{\mu}_{0,i}, \hat{\mu}_{1,i}, \hat{\sigma}_{0,i}, \hat{\sigma}_{1,i} - \hat{\pi}_{0,i-1}, \hat{\pi}_{1,i-1}, \hat{\mu}_{0,i-1}, \hat{\mu}_{1,i-1},$
$\hat{\sigma}_{0,i-1}, \hat{\sigma}_{1,i-1}| < \epsilon$ **then**
 *convergence* $\leftarrow$ *True*
 **end if**
**end while**
**return** $\hat{\pi}_{0,i}, \hat{\pi}_{1,i}, \hat{\mu}_{0,i}, \hat{\mu}_{1,i}, \hat{\sigma}_{0,i}, \hat{\sigma}_{1,i}$

Finally, calculate the *FQR* and return the highly credible quantitation results. We defined a decision rule: any quantitation result with a score higher than the threshold $t$ will be defined as a highly credible one, and *FQR* is also used to evaluate the false rate of quantitation results under the given decision rule ($FQR \leq 0.01$ by default). In this paper, the false-positive set and true-positive set fitted by EM algorithm is used to calculate *FQR*, given the threshold $t$, the calculation formula of *FQR* is as follows:

$$FQR = \frac{\pi_0 P(X > t | T = 0)}{\pi_0 P(X > t | T = 0) + \pi_1 P(X > t | T = 1)} \tag{7}$$

Where the possibility of being in the false-positive set or being in the true-positive set with a score greater than the threshold $t$ can be calculated by fitting the area of the distribution.

$$P(X > t | T = k) = \pi_k \int_t^{+\infty} f_k(x) dx, k \in \{0,1\} \tag{8}$$

**The MIR procedure**

The MIR procedure consists of three steps:

Step 1: generating the candidate glycan database and candidate glycopeptides. The maximum structure of complex and hybrid glycan types in human were converted from the glycan structures of GlycoWorkbench[44] into the canonical strings of pGlyco3[13] to form an

initialized glycan database. Then, glycans with LacNAc-structure were screened out from the initialized glycan database to construct the candidate glycan database. The glycans in candidate glycan database were grouped into 342 subnets (Supplementary Fig. 14), each of which contains glycans having the same glycan infrastructure (the term of "glycan infrastructure" we used here referring to a glycan structure excluding sialic acid units). The glycan of the identified glycopeptide was used to match all subnets in the candidate glycan database. All glycans in the matched subnet were used to produce glycopeptide candidates for the identified glycopeptide.

Step 2: extracting signals of the candidate glycopeptides. Then the glycopeptide evidence of a candidate is extracted within the preset retention time window (generated in real-time by calculating the median retention time caused by sialic acid-based bias in a spectrum, or defined by user) through the following procedure: Firstly, calculate the theoretical isotope distribution of a glycopeptide candidate **X**, find the experimental isotope peak with the closest mass within the range of ±20 ppm of the corresponding theoretical isotope peak, and form the experimental isotope distribution **Y**.

Step 3: Scoring for the candidate glycan evidence. Calculate the cosine similarity $Sim_{iso}$ between the theoretical isotope distribution **X** and the experimental isotope distribution **Y** for each mass spectrum. Rank the cosine similarity within the retention time window, if the top1 $Sim_{iso}$ is greater than 0.9, take its spectral peak as the center, extend forward and backward within the window until the similarity is <0.9. Then the start and end retention time of this evidence will be confirmed.

$$Sim_{iso} = \frac{\mathbf{X} \cdot \mathbf{Y}}{|\mathbf{X}||\mathbf{Y}|} \tag{9}$$

The area under the curve of the mono peak in the evidence is extracted as the intensity of the glycopeptide candidate. Sum up all theoretical and experimental isotope distribution $Sim_{iso}$ within the start and end retention time, which is the MIR matching score of this evidence ($Score_{MIR}$).

$$Score_{MIR} = \sum Sim_{iso}(i) \tag{10}$$

**Intensity merge strategy**

Four quantitative result files (.list), "spectra.list", "site.list", "modification.list", and "protein.list", were outputted by pGlycoQuant. The "spectra.list" gives the intensity of each glycopeptide spectrum. The "site.list" gives quantitation results on protein site-specific glycan level, which is calculated by summing up the intensity of all glycopeptide spectra identified for the same protein site-specific glycan. The "modification.list" gives quantitation results on the level of peptide sequence with specific glycan and modifications (other than glycosylation), which is calculated by summing up the intensity of all glycopeptide spectra identified for the same peptide sequence, glycan and modifications. The "protein.list" gives quantitative results on protein-level, which is calculated by summing up the intensity of all glycopeptide spectra identified for the same glycoprotein. Besides, two additional result files (.list), "glycan_occupancy.list" and "site_occupancy.list", can be outputted by pGlycoQuant. The "glycan_occupancy.list" gives the quantitative information of different glycan compositions and glycan types at a same glycosylation site. The "site_occupancy.list" gives the quantitative information of the same glycan composition at different glycosylation sites on a protein.

**Comparison of pGlycoQuant with other software tools**

To guarantee a fair comparison, we adopted the following procedure based on previously suggested rules: (1) To prevent

differences introduced by identification, pGlycoQuant also read the identification results reported by other software tools and calculates the quantitation results. (2) Missing values are analyzed, and two proportions at the protein level and protein quantitation value level are reported. (3) After the removal of the missing values, we compare the Pearson correlation and standard deviation of the intensities at the GPSM, glycopeptide, and protein levels without normalization. The key indicators are listed below (also see Supplementary Fig. 6):

PMVL (proportion of missing value in line) = the number of quantified IDs (GPSMs/glycopeptides/ proteins) with more than one missing value/the number of all quantified IDs.

PMVT (proportion of missing value in total) = the number of individual missing values/the number of all quantitation values

$$\text{Pearson correlation} = \frac{E(\mathbf{XY}) - E(\mathbf{X})E(\mathbf{Y})}{\sqrt{E(\mathbf{X}^2) - (E(\mathbf{X}))^2}\sqrt{E(\mathbf{Y}^2) - (E(\mathbf{Y}))^2}} \quad (11)$$

where $\mathbf{X}$ and $\mathbf{Y}$ are vectors of protein or peptide intensities in one run, respectively.

$$\text{Robust standard deviation} = \frac{\mathbf{R}_H - \mathbf{R}_L}{2} \quad (12)$$

where $\mathbf{R}$ is the vector of log2-transformed quantitation ratios of two runs. $\mathbf{R}_H$ and $\mathbf{R}_L$ are the 84.13% and 15.87% percentiles, respectively. This robust standard deviation was introduced in the MaxQuant paper[45]. For a normal distribution, these would be equal to each other and to the conventional definition of a standard deviation.

### Acquisition of a two-glycoproteome dataset
Proteins were extracted from human serum and fission yeast, respectively. Then, the two protein samples were treated by the same experimental procedure to obtain glycopeptides. Glycopeptides from the two samples were analyzed by LC-MS/MS, respectively. Detailed sample preparation and data acquisition methods are described in Supplementary Note 1. The data were analyzed using pGlyco3 followed by pGlycoQuant for quantitation. The detailed searching parameters are shown in Supplementary Table 3.

### Acquisition of fold change-(de)glycoproteome dataset
Proteins were extracted from human IgG, fission yeast, and human serum, respectively. Glycopeptides enriched from each sample were divided into four portions, two portions of 1 μg and two portions of 200 ng. Then, a portion of 1 μg and 200 ng glycopeptides were treated with PNGase F. Glycopeptides and deglycopeptides of each condition were analyzed by LC-MS/MS with triplicates. Detailed sample preparation and data acquisition methods are described in Supplementary Note 2. The data were analyzed using pGlyco3 followed by pGlycoQuant for quantitation. The detailed searching parameters are shown in Supplementary Table 3.

### Acquisition of three benchmark datasets
Three benchmark datasets, namely, SILAC-labeled 293T cell data, label-free HeLa cell data, and TMT-labeled 293T cell data (Supplementary Table 1), were generated with the different quantitative strategies. HeLa (TCHu187) and 293T (GNHu17) cell lines were purchased from National Collection of Authenticated Cell Cultures of China. In brief, for the SILAC-labeled 293T cell data, 293T cells were cultured in K0R0 and K6R6 media. Then, proteins were extracted from the labeled cells, mixed at a 1:1 ratio and digested. For the label-free HeLa data, HeLa cells were directly collected and used for protein extraction and digestion. For the TMT-labeled 293T cell data, proteins were extracted from

293T cells. The digests were divided into two aliquots, each of which was labeled with the TMT6plex™ label reagents TMT⁶–128 and TMT⁶–131, respectively, following the TMT6plex™ isobaric label reagent product manual (Thermo Fisher Scientific, Waltham, MA, U.S.A.), and mixed with 1:1. As with any quantitative strategies, glycopeptides were enriched from the desalted digests using ZIC-HILIC method and analyzed by LC−MS/MS. Detailed sample preparation and data acquisition methods are described in Supplementary Note 3. Three benchmark datasets were searched using different software tools (Supplementary Table 2) for quantitative performance comparison. The detailed searching parameters are shown in Supplementary Table 3.

### The proteome and N-glycoproteome data obtaining from SILAC-labeled HCC cell lines
We used the SILAC strategy to label the three cell lines MHCC97L, Hep3B and MHCCLM3 with K0R0, K4R6, and K8R10 labeling, respectively. Hep 3B (SCSP-5045) cell line was purchased from National Collection of Authenticated Cell Cultures of China. MHCC97L and MHCCLM3 cells were obtained from Liver Cancer Institute, Zhongshan Hospital of Fudan University, among which 97L and LM3 cells were established at this institute[46]. After SILAC labeling, proteins were extracted from the labeled cells, mixed at a 1:1:1 ratio and digested. The tryptic digests were then subjected to chromatographic fractionation with HILIC and used for direct proteomic quantitation and intact N-glycopeptide quantitation after ZIC-HILIC enrichment by four replicates of LC−MS/MS analysis. See Supplementary Note 3 for details. For the SILAC labeling, cells were cultured following the experimental procedure described in Supplementary Note 3 and collected after culturing for 8 generations with over 95% labeling efficiency. To confirm the performance of our SILAC labeling experiments, we mixed the proteins from different labeling cells at a 1:1:1 ratio, digested them and quantitatively analyzed them by LC−MS/MS. We used housekeeping proteins, including actin, tubulin and GAPDH, which are usually stable in organisms, as standards to evaluate the labeling efficiency. All the relative quantitative results of these housekeeping proteins showed no significant changes among the three cell lines (Supplementary Data 4), which demonstrated a fine SILAC experiment and guaranteed the feasibility of further quantitative analysis. The proteome data of SILAC-labeled HCC cell lines were analyzed using Open-pFind software[30] with open search mode for identification followed by pGlycoQuant for quantitation. The intact glycopeptide data of SILAC-labeled HCC cell lines were analyzed using pGlyco3 followed by pGlycoQuant without MIR function for quantitation with the same parameters in Supplementary Table 3.

### Benchmark, software versions
pGlycoQuant (programmed by python 3.7) supports quantitation of the identification results from Byonic, MSFragger, pFind and pGlyco services. The following search engines and quantitation engines/modes were used in this study for the pGlycoQuant-supporting quantitation test and comparison. Search engines: pGlyco3, MSFragger-Glyco, and Byonic. Quantitation engines/mode: pGlycoQuant, MSFragger-Glyco, Byologic, Skyline, and Proteome Discoverer. The detailed versions are listed in Supplementary Table 2.

### In vitro functional validation and molecular biology experiments
We utilized western blotting to detect the expression of L1CAM and FUT8 in the three cell lines Hep3B, MHCC97L, and MHCCLM3 and verify MS-based proteomic quantitative results. The 97L cells were transfected with FUT8 siRNA, pL1CAM-FLAG plasmid and pL1CAM

(N979Q)-FLAG plasmid, and the LM3 cells were knocked down by L1CAM siRNA. The effect of transfection was tested through western blot or lectin enrichment and immunoblot assays. For functional validation experiments, wound healing assays and transwell migration assays were adopted to validate the migration capacity of the above transfected cells. The invasive ability of these cells was evaluated by Matrigel invasion assay. The details of the above experiments are described in Supplementary Note 4.

## Data availability

The RAW MS data, including the two-glycoproteome dataset, the fold change-(de)glycoproteome dataset, three benchmark datasets, MIR datasets, and SILAC-labeled HCC cell lines proteome data and intact glycopeptide data, as well as the search results and the relevant analyses generated in this work have been deposited in the MassIVE repository under accession code MSV000089484. Swiss-Prot protein databases used in this study are available at UniProt (https://www.uniprot.org). Raw files of mixed-organism samples (human serum and budding yeast) containing two different proportions of the species were downloaded from the ProteomeExchange, accession code PXD023980. Detailed search parameters for all these RAW data files are listed in Supplementary Data. All the pGlyco3 result files can also be found in Supplementary Data. Source data are provided with this paper.

## Code availability

pGlycoQuant (programmed by python 3.7) is freely available on Zenodo (https://zenodo.org/record/7267832)[47] and OGP (http://www.oglyp.org/pglycoquant/). The pGlycoQuant version used in this manuscript can be downloaded from GitHub (https://github.com/Power-Quant/pGlycoQuant/releases).

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

## Acknowledgements

We thank Professor Simin He from Institute of Computing Technology, CAS, Beijing, China for his kindly directing research, providing valuable advices and moral support. We thank Xia Gao and Anqi Hu from Institutes of Biomedical Sciences, Fudan University, Shanghai, China for the help of TMT-293T cell sample preparation during the lab lockdown in the COVID-19 pandemic. This work was supported by grants from the National Natural Science Foundation of China Project (32271490 to W.C., 32171442 to C.L.), the National Key Research and Development Program (2021YFA1301602 and 2021YFA1301603 to C.L.), the Innovative Research Team of High-Level Local University in Shanghai, and the Department of Science and Technology of Henan Province, China (201400210500 to C.L.). This paper is dedicated to the memory of Professor Pengyuan Yang (1949.6.12–2021.5.31), who passed away during the paper preparation.

## Author contributions

W.C. conducted this project, performed the wet-lab experiments and data analysis, and wrote the manuscript. C.L. developed the software pGlycoQuant, performed the data analysis and revised the manuscript. S.K. conducted the wet-lab experiments, analyzed data and revised the manuscript. P.G. contribute to the software development, performed data analysis, and revised manuscript. W.Z. contributed to the pGlyco-Quant development and revised the manuscript. B.Y. conducted the wet-lab experiments for proteome and glycoproteome identification in HCC cell lines and contribute to data analysis. X.H. contributed to the pGlycoQuant development and data analysis. H.Z., G.Y., and X.Z. contribute to the LC-MS/MS analysis. Y.Z., M.L., X.Q., and M. W. contributed to the MS data analysis. W.C., C.L., and P.Y. supervised this project.

## Competing interests

The authors declare no competing interests.
