## [Peer Review File · Nature Communications]

REVIEWER COMMENTS

Reviewer #1 (Remarks to the Author):

The manuscript entitled "pGlycoQuant with a deep residual network ..." presents a post-processing software tool for re-quantification of the previously identified set of glycopeptide data. The central idea of the paper is that pGlycoQuant can reduce the rate of the missing value and therefore improve the quantitative glycoproteomics pipeline.

The revised manuscript is the next round of re-submission and has the rebuttal letter from the previous round.

The authors addressed a lot of comments raised previously and significantly revised the manuscript. A new section about "match in the run" was added to the paper. References were improved and corrected. The description of how pGlycoQuant works are now improved and the internal algorithm architecture is clearer. The strength of the pGlycoQuant is that it is open-source soft and freely available.

However, several concerns were still not properly addressed

Major comment

1) It is still unclear in the text what is special in the quantification of glycopeptides vs. peptides. In the general reply authors says: "..glycopeptide quantitation and naked peptides have some similarities.,...the core is the extraction of signal". This is not really like that. Glycopeptide quantification has several aspects:

a) Quantification of the precursor ions from the full MS scan (or reporter groups from TMT data). This is what is manuscript about for the quantification strategy and this is a pure type of proteomics strategy. Here quantification is not just "..some similarities.." it is exactly the same. the challenge is to find m/z feature and make proper retention time alignment across replicates.

b) Here, we address stoichiometry, site occupancy, and relative quantification of positional isomers (the same peptide could have either different glycans or glycan at different positions). This indeed makes the quantification strategy special and unique. But this is not the case with the manuscript. All these problems are not addressed. So the quantification strategy is around the first aspect only.

The authors said that they can capture glycopeptide evidence. This is still not clear to me how? What does it mean to capture? Recognize the glycopeptide pattern in full MS1 or just find m/z among the chemical noise? pGlycoQuant approach is based on the re-quantification of the previously identified glycopeptides (pGlyco). So it is biased to that list and glycopeptide is known. The m/z and retention time are known. So it is only necessary to perform accurate retention time alignment and recognize m/z features among chemical noise signals. But this problem has been perfectly addressed in the proteomics pipeline a lot of good solutions appeared on the market. This rise a second question

2) Authors said that they can read the identification list from the external ID software and make quantification. The reviewer asked to compare this to similar strategies available on the market (e.g. Proteome Discoverer, Progenesis, Scaffold), but it was not done. The authors only compared with Skyline (asked by another reviewer). For example, Progenesis can read mass spec data from all the most popular machines and perform spectral alignment and

feature recognition and align with your identification list. With Progenesis one could get almost 0% of missing values across replicates. Another tool (Proteome Discoverer) can work not only with Mascot and Sequest but also with MS-Fragger, Ms-Amanda, and Bionic. The last three authors forgot to comment. The reviewer made a quantitative test using Proteome Discoverer (using Minora Node) and found that PMVL was around 2% and PMVT around 4% for 4 replicates of commercial HeLa digest. For glycopeptide analysis PD with Minora demonstrated PMVL-3% and PMVT-4%.

So taking this into account pGlycoQuant improves its quantification strategy incrementally. No doubt, pGlycoQuant improves that rate but in the actual number of missing values, it is a few percent down. there is nearly nothing to improve

Taking all the above-mentioned pGlycoQuant could be another interesting quantification software among already existing powerful tools.

Comments #3-5 were properly addressed

In addition, there is a minor comment.

1) Fig.4a. What is the identification engine? Currently, it is shown only pGlycoQuant, but this is only for quantification. Perhaps should be an "identification box" in-between

2) Line 591. "...pGlycoQuant outperforms other tools in terms of precision and reproducibility". what does it mean "other tools". I guess it is only Bionic, MS-Fragger and pGlyco. Either names them or give a reference to some sort of table (where they are listed)

3) Line 760. "glycoeptide" correct to "glycopeptide"

4) I was not able to access raw data on MassIVE. Please check. Maybe it was my browser settings.

Reviewer #2 (Remarks to the Author):

The authors have done an impressive amount of work to respond to the previous review comments and I am now happy for this work to be published.

Some of the Figures are too low resolution for publication, but I assume that will be remedied in the final version.

Reviewer #3 (Remarks to the Author):

I commend the authors on their thoughtful and thorough efforts to edit the manuscript after the first round of reviews. I find the data they added from the de-glycoproteomic experiment to be compelling for their case of correct quantitation of glycopeptides, and I believe they did well to address most of my comments as well as those from the other reviewers. Although this manuscript is now much closer to a publishable form in its revised form, I still believe the authors need to address the following points before it is suitable for publication.

1. Pearson correlation in Fig. 2c is very hard to see. A way to make it bigger in the top left or bottom right corner without the "Pearson="?

2. The representation of glycoproteins, glycans, and glycosites in Fig. 4c is misleading. Why is it a pie graph? These numbers are for different values that are not subsets of a total count, so a pie graph is the incorrect display format.

3. The authors should state in the main text whether the MIR feature was used for the HCC data.

4. Why hasn't ResNet solved missing value problem for standard proteomics? Does it work for complex signals like glycopeptides better? This seems improbable. Can the authors address this in the discussion?

5. 58% of glycopeptides (6435 out of 11001) quantified in more than two replicates for the HCC cell line experiment shows me that while pGlycoQuant may have improved missing value problems in quantitative glycoproteomics experiments, it has far from solved it. As such, I believe the authors must tone down the language that claims "could largely solve the problem of random precursor selection for MS2 analysis in different replicates."

6. How can site N979 change be relevant more so than the other four N-glycosites when the abundance change due to abundance differences at the protein level? It would seem all glycosites on L1CAM would have the same effect as N979 in the mutation experiments.

7. Can the authors provide annotated spectra for the 2% of glycopeptide IDs that had the N-X-C motif? That number is close to the FDR cutoff, and it is within reason to believe they may not be accurate identifications.

8. The match in runs feature is interesting and strengthens the manuscript. However, it is the algorithm within pGlycoQuant that has the most vulnerability to false matches. The sialic acid tests they performed produced believable data that warrants inclusion in the manuscript, but these are still highly contrived experiments. I remain cautious that this would prove reliable in a sample with thousands of sialylated glycopeptides with its current implementation. Thus, I think the authors should state this even more explicitly than they have in the discussion, saying that Match In Runs is untested on truly complex mixtures and should be thoroughly evaluated manually any time it is used.

9. In the discussion, the authors should suggest the best tools to use in tandem with pGlycoQuant to manually inspect their data, as they have done throughout. Is it with a tool like Skyline? How can we in the rest of the field evaluate our data after we have used pGlycoQuant so that we can check the same quality metrics in our own data that the authors have reported here? This includes MS1 glycopeptide traces/isotope patterns and MS2 spectra.

10. Can the authors expand on the rare glycopeptides quantified in the deglycoproteomics experiment? Are they rare because they are glycopeptides whose glycans would not be

expected to be removed by PNGaseF? Reporting these glycopeptides and perhaps some annotated spectra would be helpful to understand what makes them rare.

Responses to the Reviewer's Comments

pGlycoQuant with a deep residual network for quantitative glycoproteomics at intact glycopeptide level enabling the functional exploration of site-specific glycosylation

Reviewers' Comments

Reviewer #1:

The manuscript entitled "pGlycoQuant with a deep residual network ..." presents a post-processing software tool for re-quantification of the previously identified set of glycopeptide data. The central idea of the paper is that pGlycoQuant can reduce the rate of the missing value and therefore improve the quantitative glycoproteomics pipeline. The revised manuscript is the next round of re-submission and has the rebuttal letter from the previous round.

The authors addressed a lot of comments raised previously and significantly revised the manuscript. A new section about "match in the run" was added to the paper. References were improved and corrected. The description of how pGlycoQuant works are now improved and the internal algorithm architecture is clearer. The strength of the pGlycoQuant is that it is open-source soft and freely available.

However, several concerns were still not properly addressed

Reply:

We thank the reviewer for the useful comments and pointing out the problems and putting forward suggestions, which further improved our manuscript after fixing them.

Major comment

1) It is still unclear in the text what is special in the quantification of glycopeptides vs. peptides. In the general reply authors says: "...glycopeptide quantitation and naked peptides have some similarities.....the core is the extraction of signal". This is not really like that. Glycopeptide quantification has several aspects:

a) Quantification of the precursor ions from the full MS scan (or reporter groups from TMT data). This is what is manuscript about for the quantification strategy and this is a pure type of proteomics strategy. Here quantification is not just "...some similarities.." it is exactly the same. the challenge is to find m/z feature and make proper retention time alignment across replicates.

Reply:

It is true that the quantitation strategy for the intact glycopeptides in the manuscript is a pure type of proteomics strategy (i.e., quantifying the precursor ions from the full MS scan or reporter groups from TMT data), since the main task of this paper is not to develop quantitation strategies but to develop software to support these strategies-based intact glycopeptide quantitation. We agree that the main challenge is to find m/z feature and make proper retention time alignment across replicates in both glycopeptide and naked peptide quantification. While, we think that the challenge is even bigger in glycopeptide quantification and we could barely get satisfactory results for glycopeptide quantification with traditional proteomics quantification tools. To illustrate that, we made the following analyses.

First, as the methods we described in the manuscript (Methods section “A deep-learning-based evidence matching model”), the evidence as a matrix is transformed to a 512×1 vector by the ResNet18 model, and the two vectors from two pieces of evidence are then combined into a 1024×1 vector. We extracted the 1024×1 vector for naked peptides (without glycosylation) and glycopeptides, respectively, through the ResNet18 from the “1902-Hela-Labelfree-01.raw” data and “1902-Hela-Labelfree-02.raw” data, and then we compared the value distribution (y axis) of the 1024×1 vector (x axis) between the naked peptides and glycopeptides. As shown in the **For-review Figure 1**, the scoring distribution plots varied a lot between naked peptides and glycopeptides (**For-review Figure 1a**), which reflected that glycopeptides and naked peptides had different feature distributions. In addition, different machine learning models that were trained by naked peptides (machine learning NP) and by glycopeptides (machine learning GP) were used to quantify the glycopeptides from two of the label-free data in the manuscript, “1902-Hela-Labelfree-01.raw” and “1902-Hela-Labelfree-02.raw” under the same quality control condition, respectively. As the results shown in the following **For-review Table 1**, compared with the machine learning NP method, the machine learning GP method has better quantification performance in terms of reducing missing values and quantification correlation, which also indicates to some extent that feature differences exist between naked peptides and glycopeptides and the quantification tool dedicated for glycopeptides is needed. Furthermore, after grading the vector of naked peptides in descending order, we can see that the glycopeptide features presented more significant fluctuations than that of naked peptides (**For-review Figure 1b**), which is probably related to the higher complexity

of glycopeptides. Thus, it is reasonable to infer that it is necessary to develop dedicated algorithm and software to quantify the sophisticated intact glycopeptides.

Second, the further comparison of pGlycoQuant with Proteome Discoverer, Progenesis, and Scaffold as the reviewer suggested demonstrated that compared with these available tools on the market, pGlycoQuant had better compatibility with the identification tool (**For-review Figure 3-5**), outperformed on both reducing missing values and quantitation precision (**For-review Table 5-7, Supplementary Table 4, Figure 2, Supplementary Figure 8-10**), and included an FQR algorithm for more reliable quantitation result report (**Supplementary Figure 11-13**). The detailed procedures and results of the comparison were described below for the reviewer's second question.

Third, an additional experiment conducted for the quantitation of glycopeptides with decreased concentrations further indicated the higher sensitivity of pGlycoQuant in detecting low-abundance glycopeptide signals (**For-review Figure 6, For-review Data 2**). The detailed experimental procedures and the corresponding results were also shown below in the reply for the reviewer's second question

Last, we want to emphasize that in addition to addressing issues of finding an evidence in its original run and in the parallel run to reduce missing values, measuring whether this evidence is the true positive one or random noises is also an important issue that is exactly what pGlycoQuant addressed in the last revision according to the reviewer's suggestion by proposing a false quantitation rate (FQR) estimation method based on the matching scores given by matching model and a mixture distribution model fitted by the Expectation-Maximization (EM) algorithm for ruling out false quantitative results with 1% FQR (**Figure 1d, Supplementary Note 2, Supplementary Figure 3-5**).

The above points help us to understand that the complexity of glycopeptides and the differences between the glycopeptides and the naked peptides make the intact glycopeptides hard to be accurately quantify by proteome quantification software; thus, the dedicated software with well-designed fundamental data structures are needed. We hope that our pGlycoQuant could provide out-of-the-box functionality in intact glycopeptide quantitation with largely improved quantitation performance, reliable quality control, and widely application potentials (such as MIR function and the newly-added functions for the study of glycosylation heterogeneity that were described in the following aspect "b").

For-review Figure 1 The value distribution of inputted 1024×1 vector. The x axis represents the 1024×1 vector, and the y axis represents the value of vector. (a) Distribution of 1024 points of 1024×1 vector in default order. (b) Grading the vector of naked peptides in descending order.

For-review Table 1 The performance of machine learning GP and machine learning NP on glycopeptide quantitation in terms of reducing missing values and quantification correlation.

		machine learning GP for glycopeptide quantitation	machine learning NP for glycopeptide quantitation
PSM	PMVT	3.66%	17.74%
	PMVL	7.32%	35.48%
	Pearson	0.992	0.801
	Std.	0.145	1.176
Peptide	PMVT	2.19%	9.00%
	PMVL	4.37%	18.01%
	Pearson	0.989	0.960
	Std.	0.178	0.558
Protein	PMVT	3.22%	7.89%
	PMVL	6.43%	15.89%
	Pearson	0.991	0.973
	Std.	0.16	0.395

Note: machine learning NP refers to the machine learning model that was trained by naked peptides and machine learning GP refers to the machine learning model that was trained by glycopeptides.

b) Here, we address stoichiometry, site occupancy, and relative quantification of positional isomers (the same peptide could have either different glycans or glycan at different positions). This indeed makes the quantification strategy special and unique. But this is not the case with the manuscript. All these problems are not addressed. So the quantification strategy is around the first aspect only.

Reply:

We have carefully considered the issues raised by the reviewer and we agree that the stoichiometry, site occupancy, and relative quantification of positional isomers are indeed very important aspects in the research of glycosylation. Admittedly, they are not of primary issues that pGlycoQuant focused on and solved. Still, we tried to propose some solutions as much as possible at current stage. Thus, we added two new functions to the pGlycoQuant, aiming at analysis of “the same peptide could have either different glycans or glycan at different positions” in the revised version. One is for calculating the intensity of different glycans at a same protein glycosylation site and the other is for calculating the intensity of the same glycan at different glycosylation sites on a protein. Correspondingly, two additional result files (.list), “glycan_occupancy.list” and “site_occupancy.list”, are reported and outputted by pGlycoQuant.

The “glycan_occupancy.list” gives the quantitative information of different glycan compositions and glycan types at a same glycosylation site according to the following equation (**Formula 1, 2**). We took a screenshot of the result list from the quantitation results of label-free HeLa cell data (**For-review Table 2 and 3**). As shown in **For-review Table 2**, column C lists all the glycan compositions at a protein glycosylation site, and column D corresponds to the quantitation ratio of each glycan to the total intensity of this glycosylation site (**Formula 1**). Column E to H shows the quantitation ratio of five glycan types, oligomannose, complex/hybrid, sialylation, fucosylation, and M6P, respectively (**Formula 2**). The grouping rules for the five categories were noted at the bottom the list file.

The “site_occupancy.list” gives the quantitative information of the same glycan composition at different glycosylation sites on a protein according to the following equation (**Formula 3**). As shown in **For-review Table 3**, column C lists all the glycosylation sites of a glycoprotein, which were modified by a certain glycan composition, and column D corresponds to the quantitation ratio of each site that is modified by a certain glycan to the total intensity of this glycoprotein (**Formula 3**).

With the above results, we can freely calculate the intensity of a certain glycan composition or glycan types on a same glycosylation site (e.g. multiplying the ratio in column D-I by the intensity value in column K-S in **For-review Table 2**) or calculate the intensity of a glycosylation site modified by a certain glycan (e.g. multiplying the ratio in column D by the intensity value in column F-N in **For-review Table 3**) according to our need. In addition, we could easily obtain the glycosylation heterogeneity of any glycoprotein that we are interested in based on the quantitation results. For example, the glycan quantitative distribution at each glycosite of the lysosome-associated membrane glycoprotein 1 (LAMP-1) from the label-free HeLa cell data (**For-review Data 1**) was shown in **For-review Figure 2a**, from which we can see that hybrid/complex glycan type only occurred at the glycosite 76, 84 and 249, while glycosite 322 only contains oligomannose glycan type. Another example about the glycosite quantitative distribution with a certain glycan composition of the prolow-density lipoprotein receptor-related protein 1 (LRP-1) was shown in the **For-review Figure 2b**, from which we can see that the high-mannose glycan type H8N2 was widely distributed at all the nine glycosites of LRP-1, and variety of glycan compositions were observed at glycosite 1511 and 1763.

Though some issues, such as the quantification of isomers, which requires the development of dedicated strategies, are currently not addressed in this work, we hope that the two newly added functions in the pGlycoQuant as described above could provide some solutions for the study of the glycosylation heterogeneity. And we would like to thank for the reviewer 's comments, which provide us with more thoughts and directions for the further development of a better quantification tool.

$$R_{p,s,gc} = \frac{\sum \text{Intensity}_{p,s,gc}}{\sum \text{Intensity}_{p,s}} \quad (\text{Formula 1})$$

$$R_{p,s,gt} = \frac{\sum \text{Intensity}_{p,s,gt}}{\sum \text{Intensity}_{p,s}} \quad (\text{Formula 2})$$

$$R_{p,gc,s} = \frac{\sum \text{Intensity}_{p,gc,s}}{\sum \text{Intensity}_{p,gc}} \quad (\text{Formula 3})$$

Formula 1-3, where p is one protein type, s is one site position, gc represents one glycan composition and gt represents the glycan type. Σ represents the sum of all samples.

For-review Table 2 A screenshot of “glycan_occupancy.list” from the quantitation results of label-free HeLa cell data.

#	A	B	C	D	E	F	G	H	I	J	K	L	M	N	O	P	Q	R	S
1	Protein	Site	Site_Specific_Glycan(H.N.A.F)	Glycan_Ratio	Oligosaccharide	Complex/Hybrid	Sialylation	Fucosylation	MPEP	Er 1902	Hela 1902	Hela 1902	Hela 1902	Hela 1902	Hela 1902	Hela 1902	Hela 1902	Hela 1902	Hela 1902
1	pp561997TAL_HUMAN	460	[8.2.0.0][9.2.0.0][7.2.0.0][10.2.0.0]	1.975e-01 7.923e-01 9.841e-01 3.360e-04	1.00e+00	-	-	-	-	-	-	-	-	-	-	-	-	-	-

For-review Table 3 A screenshot of “site_occupancy.list” from the quantitation results of label-free HeLa cell data.

#	A	B	C	D	E	F	G	H	I	J	K	L	M	N	
1	Protein	Glycan(H.N.A.F)	Glycan_Specific_Sites	Site_ratio	Empty	1902	Hela	1902	Hela	1902	Hela	1902	Hela	1902	Hela
1	pp561997TAL_HUMAN	[8.2.0.0]	460:113:18383:840:1102	3.127e-01 6.375e-02 1.942e-01 3.001e-01 9.956e-02 2.938e-02	4.7e+10	4.75e+10	4.67e+10	5.50e+10	4.68e+10	4.65e+10	4.8e+10	4.77e+10	4.66e+10	4.66e+10	

For-review Figure 2 The statistic information from “glycan_occupancy.list” and “site_occupancy.list”. (a) Glycan composition and glycan type statistics at each site of lysosome-associated membrane glycoprotein 1 (LAMP1) through the “glycan_occupancy.list”. (b) Glycosite occupancy statistics with each glycan composition of prolow-density lipoprotein receptor-related protein 1 (LRP1) through the “site_occupancy.list”.

The authors said that they can capture glycopeptide evidence. This is still not clear to me how? What does it mean to capture? Recognize the glycopeptide pattern in full MS1 or just find m/z among the chemical noise?

Reply:

Thanks for your comments. We are sorry for the confused description. The “capture glycopeptide” actually referred to the process of obtaining the quantitative values of the glycopeptides, which includes the extracting glycopeptide signals and quantitation processing. We think the “capture glycopeptide evidence” we previously used in the manuscript was not expressed accurately. Thus, we have modified the expression throughout the manuscript accordingly and have revised and added the details of this process in the “Workflow of pGlycoQuant (step 2 and step 3)” of the Methods section. Briefly, the reconstructed “isotopic chromatograms” are named as “evidence” in this

paper. For each input GPSM, pGlycoQuant calculates the theoretical distribution of isotopic peaks using a stepwise convolution algorithm m/z and identifies experimental isotopic peaks in a range of full MS scans where the glycopeptide may be expected. For each MS scan, pGlycoQuant applies a ppm-level m/z tolerance window (normally $\pm 10 \sim \pm 20$ ppm, can be defined by users) around the theoretical m/z values of the isotopic peaks of the glycopeptide to select the experimental peaks. These experimental intensities along the retention time axis in contiguous MS scans are assembled into a chromatogram. This chromatogram extends both left and right from the trigger MS scan until the intensity drops below 10% of the apex of the extending profile. Then, in MBR, for each identified glycopeptide in one run, pGlycoQuant detects the corresponding evidence in other runs, i.e., matches evidence between runs. Given an identified glycopeptide in one run, pGlycoQuant constructs its evidence (termed reference evidence) in the full MS scans and then calculates the matching scores of all the evidences (termed candidate evidence) with the same precursor mass in a ± 2 minutes retention time window (default, or can be defined by users) in another run. From the start to the end of the retention time window, pGlycoQuant enumerates the isotopic chromatograms as the candidate evidence, and calculates the matching scores between reference evidence and each candidate evidence. This matching score of two pieces of evidence is calculated based on the same well-trained deep-learning-based evidence matching model and the pair of pieces of evidence with the maximum matching score is selected to obtain the quantitation result between different runs. Finally, the quantitative quality control was automatically performed to estimate the false quantitation rate (FQR) of the quantitation results. In MIR, pGlycoQuant constructs glycopeptide evidences in full MS scans for the glycopeptide candidates that were produced from the identified glycopeptides within the preset retention time window through comparison the isotope distribution of the theoretical one with the experimental one within a run.

pGlycoQuant approach is based on the re-quantification of the previously identified glycopeptides (pGlyco). So it is biased to that list and glycopeptide is known. The m/z and retention time are known. So it is only necessary to perform accurate retention time alignment and recognize m/z features among chemical noise signals. But this problem has been perfectly addressed in the proteomics pipeline a lot of good solutions appeared on the market. This rise a second question.

2) Authors said that they can read the identification list from the external ID software and make quantification. The reviewer asked to compare this to similar strategies available on the market (e.g. Proteome Discoverer, Progenesis, Scaffold), but it was not done. The authors only compared with Skyline (asked by another reviewer). For example, Progenesis can read mass spec data from all the most popular machines and perform spectral alignment and feature recognition and align with your identification list. With Progenesis one could get almost 0% of missing values across replicates. Another tool (Proteome Discoverer) can work not only with Mascot and Sequest but also with MS-Fragger, Ms-Amanda, and Bionic. The last three authors forgot to comment. The reviewer made a quantitative test using Proteome Discoverer (using Minora Node) and found that PMVL was around 2% and PMVT around 4% for 4 replicates of commercial HeLa digest. For glycopeptide analysis PD with Minora demonstrated PMVL-3% and PMVT-4%.

So taking this into account pGlycoQuant improves its quantification strategy incrementally. No doubt, pGlycoQuant improves that rate but in the actual number of missing values, it is a few percent down. there is nearly nothing to improve

Reply:

Thanks for the reviewer's comments. As we have illustrated above, the features of glycopeptides showed more significant fluctuations than that of naked peptides (**For-review Figure 1**), which is probably related to the high complexity of glycopeptides and requires the development of the dedicated software tool for glycopeptide quantification. Still, we agree that a comprehensive comparison of pGlycoQuant with available tools on the market is necessary and we are sorry for the omission of comparison with some widely approved proteomics software tools in the last revision. Accordingly, we made a further comparison as the reviewer suggested in this revision.

First of all, in addition to the MSFragger-Glyco, Byologic and Skyline, we tried to compare the pGlycoQuant with Proteome Discoverer (PD), Progenesis, and Scaffold (**For-review Table 4**) on the three benchmark datasets, including SILAC-labeled 293T cell data, label-free HeLa cell data, and TMT-labeled 293T cell data (**Supplementary Table 1**). In the process of comparison, PD embedded with Byonic node worked well, while the other two software have many problems in operation. For the use of Scaffold, since there is no introduction on the Scaffold website (<https://support.proteomesoftware.com/hc/en-us>) about how to use it for glycopeptide quantification, we consulted with the technical support of Scaffold and got the replies

that they do not think it might be the best bet and suggested us to try loading data from Byonic run in PD versions 2.0 and 2.1 if we have to do so (**For-review Figure 3a**). Unfortunately, our PD version is 3.0, and the version 2.1 that we got from the technical support stopped online activation and couldn't be activated with the help of technical support (**For-review Figure 3b**). For the use of Progenesis, the official website only provides a plug-in for the import of protein identifications generated by Byonic (**For-review Figure 4a**). We followed the steps to download the plugin and tried to use it with Progenesis-QI for Proteomics (Progenesis-QIP) to quantify intact glycopeptide identifications from Byonic; however, even with the help of technical support, it still did not work (**For-review Figure 4b**). Overall, neither Scaffold nor Progenesis has a mature and automated process for quantifying intact glycopeptides. Therefore, we do not think the two software tools to be comparable with our pGlycoQuant in terms of the compatibility with glycopeptide quantification.

Nevertheless, in order to implement the glycopeptide quantification by Scaffold and Progenesis, we tried to go through the process of identification and quantification with a semi-manual approach. As shown in the **For-review Figure 5a**, to quantify glycopeptides with the Scaffold, we first imported the Raw data (.raw) into Byonic for glycopeptide identification and obtained the identification results (Output.spectra) as well as the MGF and MZID files (.mgf and .mzid) that were generated in the process of identification. Then, Scaffold loaded the MGF and MZID files and reported the intensity values of all the PSMs (psm.csv). Finally, we manually compared the identification results from Byonic "Output.spectra" with the intensity values of all the PSMs "psm.csv" reported by Scaffold. As shown in the **For-review Figure 5b**, to quantify glycopeptides with the Progenesis, we first imported the Raw data (.raw) into Progenesis-QIP to obtain intensity values of all the precursor ions (pep-ion.csv) and the MGF file (.mgf), respectively. Then, we used Byonic to identify the MGF file (.mgf) generated by Progenesis to get the glycopeptide identification results (Output.spectra). And finally, we manually compared the identification results from the Byonic "Output.spectra" with the intensity values of all the precursor ions from the Progenesis-QIP "pep-ion.csv" through matching the retention time and the m/z values to get the quantitation results on glycopeptide level.

Through the above semi-manual approach, we could partially get the quantitation results from the three benchmark datasets using Scaffold and Progenesis. However, the results are not ideal. As shown in the **For-review Table 5-7**, Progenesis could only be

used for label-free glycopeptide quantitation and reported suboptimal results for missing values (73.18% PMVL and 73.18% PMVT). Although Scaffold is capable for all the three types of datasets, its performance is unsatisfactory on both reducing missing values and quantitation precision, especially for the label-free and SILAC-based glycopeptide quantitation (**For-review Table 5 and 6**).

PD embedded with Byonic node can process glycopeptide quantitation unimpededly, though it can only directly report the quantitation results on glycopeptide level but not the GPSMs and glycoprotein level. Considering the incompatibility and unsatisfactory results of Scaffold and Progenesis for glycopeptide quantitation as described above, we did not add the comparison of Scaffold and Progenesis but only a comparison of the PD in the manuscript, since such a comparison is not fair for the two software tools that are dedicated for proteome quantitation and barely used for intact glycopeptide quantitation.

In the revision, we have added the comparison of pGlycoQuant with PD on glycopeptide quantitation using the three benchmark datasets, including SILAC-labeled 293T cell data, label-free HeLa cell data, and TMT-labeled 293T cell data, as well as the mixed-organism sample data and the fold change-(de)glycoproteome data (**Figure 2a, Supplementary Table 1-3**). The results showed that pGlycoQuant outperforms all the other software tools including PD (**Supplementary Table 2**) both in the aspect of reducing missing values and increasing quantitative precision (**Figure 2b, Supplementary Table 4**). For example, PD reported missing values of 20.89% PMVL and 13.56% PMVT, and 31.99% PMVL and 31.99% PMVT for the SILAC and TMT-based quantitation, respectively. In contrast, pGlycoQuant only reported 1.47% PMVL and 0.93% PMVT, and 0.27% PMVL and 0.27% PMVT, respectively, when using the same identification tool Byonic, which were even better when using the pGlyco3-pGlycoQuant portfolio (0.73% PMVL and 0.36% PMVT, and 0.16% PMVL and 0.16% PMVT for SILAC and TMT-based quantitation, respectively) (**Supplementary Table 4**). For the label-free quantitation, the missing value problem was ameliorated in PD reporting 10.76% PMVL and 4.89% PMVT, which was still reduced by 19% in PMVL and 30% in PMVT by pGlycoQuant. The quantitative results reported by pGlycoQuant also achieved higher correlation and lower standard deviation (**Figure 2c, Supplementary Figure 8-Supplementary Figure 10**), as well as better quantification precision by accessing the precision of quantification of the mix-organism sample data (**Figure 2d**) than other tools including PD. The detailed information about the

parameters of the PD and the corresponding results for the three benchmark datasets were added in the **Supplementary Table 3 and Supplementary Table 4**.

It was noted that although PD performed fine in terms of missing values for the label-free glycopeptide quantitation, it showed unconvincing results for the fold change-(de)glycoproteome experiment on the basis of how often the software is actually quantifying glycopeptide IDs vs background noises. (**Supplementary Note 2, Supplementary Figure 11-Supplementary Figure 13**). As the results shown in **Supplementary Figure 11-Supplementary Figure 13**, PD could quantify a high ratio of glycopeptides in deglycosylation samples either in simple or complex samples, which were shown to be the false-positive cases reported by PD and indicated the lack of quality control of PD for intact glycopeptide quantitation. That is what pGlycoQuant solved in the last revision according to the reviewer's suggestion by proposing a false quantitation rate (FQR) estimation method based on the matching scores given by matching model and a mixture distribution model fitted by the Expectation-Maximization (EM) algorithm for ruling out false quantitative results with 1% FQR (**Figure 1c, Supplementary Figure 3, and "Method" section**).

In addition, we conducted an experiment for the quantitation of glycopeptides with the decreased concentrations to further investigate the sensitivity of pGlycoQuant, Byologic, and the three software the reviewer suggested in detecting low-abundance glycopeptides signals. The experiment workflow was shown in the following **For-review Figure 6a**. First, different amounts of protein digests from the three standard glycoproteins, IgG, HPT, and Fetuin were mixed with 250 ng *E. coli* protein digests as background, respectively. Then, the mixtures were analyzed by LC-MS/MS. Finally, the data were identified by Byonic and quantified by different software tool, including pGlycoQuant, Byologic, PD, Progenesis, and Scaffold, respectively. We statistically analyzed the missing values reported by different software for glycopeptide quantitation of different initial amounts of glycoproteins (quantitation results provided in **For-review Data 2**). As the results shown in the following **For-review Figure 6b**, pGlycoQuant only reported 1.57% PMVT for the glycopeptide quantitation of a relatively high initial amounts of glycoproteins (400 ng) and can still reported 16.54% PMVT when the initial amount reduced to 50 ng. In contrast, the PMVT reported by the other four tools were among 10%-50% for initial amounts of 400 ng glycoproteins and were even worse, up to over 50% for the low initial amounts of 50ng glycoproteins.

We believe that the above analysis and comparison could further demonstrate the advantages of pGlycoQuant and address the reviewer’s concerns. The related data have been added in the manuscript or provided as For-review materials, and the corresponding contents have been added into the manuscript.

For-review Table 4 Sources of Software tools.

Software portfolio	Identification software	Version of identification software	Quantification software	Version of quantitation software
Byonic+Byologic	Byonic	V4.0.1	Byologic	V4.0.1
Byonic+PD	Byonic	V2.16.11	PD	3.0
Byonic+pGlycoQuant	Byonic	V4.0.1	pGlycoQuant	v1.1_build20220920
MSFragger-Glyco+ MSFragger-Glyco	MSFragger-Glyco	FragPipe15.0, MSFragger_3.2, philosopher_v3.3.12 (SILAC-labeled 293T cell data and label-free HeLa cell data)	MSFragger-Glyco	FragPipe15.0, MSFragger_3.2, philosopher_v3.3.12 (SILAC-labeled 293T cell data and label-free HeLa cell data)
		FragPipe16.0, MSFragger_3.3, philosopher_v4.0.0 (TMT-labeled 293T cell data)		FragPipe16.0, MSFragger_3.3, philosopher_v4.0.0 (TMT-labeled 293T cell data)
MSFragger-Glyco+ pGlycoQuant	MSFragger-Glyco	FragPipe15.0, MSFragger_3.2, philosopher_v3.3.12 (SILAC-labeled 293T cell data and label-free HeLa cell data)	pGlycoQuant	v1.1_build20220920
		FragPipe16.0, MSFragger_3.3, philosopher_v4.0.0 (TMT-labeled 293T cell data)		
pGlyco3+Skyline	pGlyco3	pGlyco3	Skyline	21.2.1.455
pGlyco3+ pGlycoQuant	pGlyco3	pGlyco3	pGlycoQuant	v1.1_build20220920
Byonic+Progenesis	Byonic	V2.16.11	Progenesis QI for proteomics	V4.2

			(Progenesis-QIP)	
Byonic+Scaffold	Byonic	V2.16.11	Scaffold Q+ / Scaffold Quant	4.0.5 / 5.0.3

For-review Figure 3 The difficulties we met during the use of Scaffold for intact glycopeptide quantitation. (a) The replies from the technical support of Scaffold. (b) The failure process of PD 2.1 installation.

a

b

For-review Figure 4 The difficulties we met during the use of Progenesis for intact glycopeptide quantitation. (a) The introduction of the plugin for the import of protein identifications generated by Byonic on the Progenesis website. (b) The screenshot of the Hint Window showing the failure in importing search results after installation of the plugin under the direction of the technical support.

a Support for Byonic

Download

About this plug-in

Supports the import of protein identifications generated by Protein Metrics Byonic™ software.

To identify using Byonic™, follow these steps:

1. In Progenesis QI for proteomics, at the **Identify Peptides** screen, make sure the **Limit fragment ion count** checkbox is unchecked, and click the **Export N ms/ms spectra** button.
2. Save the export to a .mgf file.
3. Import the .mgf file into Byonic™.
4. Export identifications from Byonic™, in mzIdentML format (.mzid).
5. Import the resulting .mzid file into Progenesis QI for proteomics.

b

For-review Figure 5 The semi-manual approaches we used for the glycopeptide quantification by Scaffold (a) and Progenesis (b).

For-review Table 5 Quantitation results of the Label-free HeLa cell data.

Dataset	Label-free HeLa cell data								
Identification	Byonic	Byonic	Byonic	MSFragger-Glyco	MSFragger-Glyco	pGlyco3	pGlyco3	Byonic	Byonic
Quantitation	Byologic	PD	pGlycoQuant	MSFragger-Glyco	pGlycoQuant	Skyline	pGlycoQuant	Progenesis	Scaffold
Number of GPSMs	N/A	N/A	31491	N/A	7849	N/A	26288	N/A	N/A
PMVL (%)	N/A	N/A	13.55%	N/A	13.98%	N/A	11.33%	N/A	N/A
PMVT (%)	N/A	N/A	4.07%	N/A	3.57%	N/A	2.66%	N/A	N/A
Pearson	N/A	N/A	0.983-0.995	N/A	0.984-0.994	N/A	0.986-0.995	N/A	N/A
Standard deviation	N/A	N/A	0.129-0.308	N/A	0.136-0.298	N/A	0.124-0.296	N/A	N/A
Number of glycopeptides*	2716	2564	2740	607*	1651	1931/2348 [#]	2348	735/2740 ^{&}	2447
PMVL (%)	61.45%	10.76%	8.72%	27.68%	8.96%	18.19%	6.94%	0%/73.18%	66.12%
PMVT (%)	30.12%	4.89%	3.42%	13.45%	2.43%	17.81%	2.03%	0%/73.18%	36.96%

Pearson	0.941-0.973	0.955-0.980	0.974-0.995	0.940-0.987	0.976-0.994	0.877-0.961	0.979-0.996	0.938-0.993	0.899-0.946
Standard deviation	0.186-0.383	0.212-0.372	0.179-0.386	0.159-0.393	0.166-0.334	0.231-0.407	0.172-0.359	0.147-0.348	0.375-0.543
Number of glycoproteins	N/A	N/A	405	333	362	N/A	483	N/A	N/A
PMVL (%)	N/A	N/A	7.41%	37.24%	9.12%	N/A	9.73%	N/A	N/A
PMVT (%)	N/A	N/A	3.95%	27.83%	2.92%	N/A	2.48%	N/A	N/A
Pearson	N/A	N/A	0.983-0.998	0.971-0.998	0.986-0.997	N/A	0.991-0.998	N/A	N/A
Standard deviation	N/A	N/A	0.120-0.301	0.068-0.244	0.121-0.297	N/A	0.123-0.304	N/A	N/A

* MSFragger-Glyco reports quantitation results of glycopeptides at peptide sequence level, other software report at modified glycopeptide level.

The number after the slash indicates the number of input entries for quantitation by Skyline, the number before the slash indicates the number of quantitation results reported by Skyline.

& The number after the slash indicates the number of input entries for quantitation by Progenesis, the number before the slash indicates the number of quantitation results reported by Progenesis.

The N/A indicates the value is not reported by the software.

For-review Table 6 Quantitation results of the SILAC-labeled 293T cell data.

Dataset	SILAC-labeled 293T cell data								
Identification	Byonic	Byonic	Byonic	MSFragger-Glyco	MSFragger-Glyco	pGlyco3	pGylco3	Byonic	Byonic
Quantitation	Byologic	PD	pGlycoQuant	MSFragger-Glyco	pGlycoQuant	Skyline	pGlycoQuant	Progenesis	Scaffold
Number of GPSMs	N/A	N/A	5389	N/A	1553	N/A	4492	N/A	N/A
PMVL (%)	N/A	N/A	1.15%	N/A	2.51%	N/A	0.31%	N/A	N/A
PMVT (%)	N/A	N/A	0.68%	N/A	1.26%	N/A	0.16%	N/A	N/A
Pearson	N/A	N/A	0.830	N/A	0.865	N/A	0.831	N/A	N/A

Standard deviation	N/A	N/A	0.982	N/A	1.001	N/A	0.940	N/A	N/A
Number of glycopeptides	985	627	1021	361*	523	807/827#	827	N/A	979
PMVL (%)	53.40%	20.89%	1.47%	11.08%	0.57%	2.54%	0.73%	N/A	63.02%
PMVT (%)	26.75%	13.56%	0.93%	N/A	0.29%	2.48%	0.36%	N/A	31.51%
Pearson	0.665	0.793	0.879	N/A	0.892	0.649	0.863	N/A	0.521
Standard deviation	1.693	0.957	1.077	0.975	1.042	1.037	1.11	N/A	1.658
Number of glycoproteins	N/A	N/A	253	148	181	N/A	252	N/A	N/A
PMVL (%)	N/A	N/A	1.98%	4.73%	0.55%	N/A	1.19%	N/A	N/A
PMVT (%)	N/A	N/A	1.19%	N/A	0.28%	N/A	0.60%	N/A	N/A
Pearson	N/A	N/A	0.878	N/A	0.92	N/A	0.882	N/A	N/A
Standard deviation	N/A	N/A	0.945	0.769	0.735	N/A	1.080	N/A	N/A

* MSFragger-Glyco reports quantitation results of glycopeptides at peptide sequence level, other software report at modified glycopeptide level.

The number after the slash indicates the number of input entries for quantitation by Skyline, the number before the slash indicates the number of quantitation results reported by Skyline.

The N/A indicates the value is not reported by the software. MSFragger-Glyco reports the ratios of light/heave intensity instead of intensity values, thus the PMVL and PMVT values and Pearson correlation coefficient cannot be calculated. Progenesis does not support quantitation of SILAC data.

For-review Table 7 Quantitation results for the TMT-labeled 293T cell data.

Dataset	TMT-labeled 293T cell data								
Identification	Byonic	Byonic	Byonic	MSFragger-Glyco	MSFragger-Glyco	pGlyco3	pGlyco3	Byonic	Byonic
Quantitation	Byologic	PD	pGlycoQuant	MSFragger-Glyco	pGlycoQuant	Skyline	pGlycoQuant	Progenesis	Scaffold

Number of GPSMs	N/A	N/A	857	1463	1461	N/A	1370	N/A	N/A
PMVL (%)	N/A	N/A	0.82%	0.68%	0.75%	N/A	0.88%	N/A	N/A
PMVT (%)	N/A	N/A	0.76%	0.55%	0.58%	N/A	0.69%	N/A	N/A
Pearson	N/A	N/A	0.983	0.983	0.983	N/A	0.972	N/A	N/A
Standard deviation	N/A	N/A	0.133	0.127	0.127	N/A	0.123	N/A	N/A
Number of glycopeptides	N/A	322	372	332*	670	630/636 [#]	636	N/A	147
PMVL (%)	N/A	31.99%	0.27%	4.52%	0.15%	11.64%	0.16%	N/A	6.80%
PMVT (%)	N/A	31.99%	0.27%	4.52%	0.07%	11.64%	0.16%	N/A	3.40%
Pearson	N/A	0.977	0.995	0.999	0.996	0.997	0.990	N/A	0.778
Standard deviation	N/A	0.112	0.105	0.078	0.099	0.068	0.094	N/A	0.117
Number of glycoproteins	N/A	N/A	117	205	212	N/A	169	N/A	N/A
PMVL (%)	N/A	N/A	0.00%	3.90%	0.00%	N/A	0.00%	N/A	N/A
PMVT (%)	N/A	N/A	0.00%	3.90%	0.00%	N/A	0.00%	N/A	N/A
Pearson	N/A	N/A	0.998	0.999	0.999	N/A	0.999	N/A	N/A
Standard deviation	N/A	N/A	0.092	0.07	0.070	N/A	0.063	N/A	N/A

* MSFragger-Glyco reports quantitation results of glycopeptides at peptide sequence level, other software report at modified glycopeptide level.

[#] The number after the slash indicates the number of input entries for quantitation by Skyline, the number before the slash indicates the number of quantitation results reported by Skyline.

The N/A indicates the value is not reported by the software. Progenesis does not support quantitation of TMT labeling data.

For-review Figure 6 The experiment of quantifying glycopeptides with the decreased concentrations for the further investigation of the quantitative sensitivity of pGlycoQuant. (a) The experimental workflow. (b) A bar chart of the missing values as PMVT reported by different software for the glycopeptide quantitation of different initial amounts of glycoproteins.

Taking all the above-mentioned pGlycoQuant could be another interesting quantification software among already existing powerful tools.

Reply:

Thanks for the reviewer's very constructive suggestions, which further improved our manuscript after fixing them and bring us more perspectives in the future development.

Comments #3-5 were properly addressed

In addition, there is a minor comment.

1) Fig.4a. What is the identification engine? Currently, it is shown only pGlycoQuant, but this is only for quantification. Perhaps should be an "identification box" in-between

Reply:

Thanks for the careful review. We have added the search engine of Open-pFind for proteome identification and pGlyco3 for intact glycopeptide identification in Figure 4a.

2) Line 591. "...pGlycoQuant outperforms other tools in terms of precision and reproducibility". what does it mean "other tools". I guess it is only Bionic, MS-Fragger and pGlyco. Either names them or give a reference to some sort of table (where they are listed)

Reply:

Thanks for your suggestion. We have added a reference to the Supplementary Table 2 (where the software tools for comparison are listed).

3) Line 760. "glycoepptide" correct to "glycopeptide"

Reply:

We have corrected it.

4) I was not able to access raw data on MassIVE. Please check. Maybe it was my browser settings.

Reply:

We recommend reviewers to utilize an FTP client (such as WinSCP or FileZilla) to access raw data if the reviewer want to download the data. Maybe, the files are too large to download using a web browser. The website of MassIVE can be used to check file types and numbers. We have tried to use different browsers (including Chrome, Internet Explorer and Firefox) from different IP addresses in China or in America to view the list of data from MassIVE, all of which worked.

Here are steps when downloading by FTP client and hope it helps:

1) Copy FTP download link (ftp://MSV000089484@massive.ucsd.edu) to Host bar and password is a. Click the Quickconnect button and log in to the MassIVE database.

2) After connecting to MassIVE, you can choose the filename of "raw" to download the raw data.

Reviewer #2:

The authors have done an impressive amount of work to respond to the previous review comments and I am now happy for this work to be published.

Some of the Figures are too low resolution for publication, but I assume that will be remedied in the final version.

Reply:

We thank the reviewer again for the approval of pGlycoQuant. We have renewed the Figures with high resolution. All the figures would be remedied in the final version.

Reviewer #3:

I commend the authors on their thoughtful and thorough efforts to edit the manuscript after the first round of reviews. I find the data they added from the de-glycoproteomic experiment to be compelling for their case of correct quantitation of glycopeptides, and I believe they did well to address most of my comments as well as those from the other reviewers. Although this manuscript is now much closer to a publishable form in its revised form, I still believe the authors need to address the following points before it is suitable for publication.

Reply:

We thank the reviewer for the approval of our revision work and the further suggestions. We have tried to address the following points to further improve our manuscript.

1. Pearson correlation in Fig. 2c is very hard to see. A way to make it bigger in the top left or bottom right corner without the “Pearson=”?

Reply:

We have modified it accordingly.

2. The representation of glycoproteins, glycans, and glycosites in Fig. 4c is misleading. Why is it a pie graph? These numbers are for different values that are not subsets of a total count, so a pie graph is the incorrect display format.

Reply:

We have modified the Figure 4c and provided a bar graph alternatively.

3. The authors should state in the main text whether the MIR feature was used for the HCC data.

Reply:

Thanks for your suggestion. MIR was not used for the HCC data. We have stated it in the Methods part “The proteome and N-glycoproteome data obtaining from SILIAC-labeled HCC cell lines”.

4. Why hasn't ResNet solved missing value problem for standard proteomics? Does it work for complex signals like glycopeptides better? This seems improbable. Can the authors address this in the discussion?

Reply:

Thanks for the reviewer's question. At present, there are indeed very few studies for quantitative proteomics using ResNet machine learning, since acceptable quantification performance can already be obtained for standard proteomics without ResNet [1-4]. However, intact glycopeptide quantitation suffers from impaired accuracy and large numbers of quantitative missing values. Hence, we developed pGlycoQuant. At the beginning, we used conventional similarity scoring method (calculating the cosine similarity of two evidences), but this method could not guarantee adequate distinction between true and false positive evidence matches. We think that the superiority of ResNet in solving the image recognition [5-7] makes it has great potential in accurate recognition of chromatographic curves and could also be used for proteomic quantification to solve missing value problem, but only if the dedicated model is trained. To briefly illustrate that, we used conventional similarity scoring method (calculating the cosine similarity) and ResNet machine learning method to quantify the naked peptides identified by pFind [8] and glycopeptides identified by pGlyco3 [9] from two of the label-free data in the manuscript, “1902-Hela-Labelfree-01.raw” and “1902-Hela-Labelfree-02.raw” under the same quality control condition. As the results shown in the following **For-review Table 8**, the machine learning method has better performance in terms of reducing missing values for both glycopeptides and naked peptides than the cosine similarity method when using dedicated machine learning model that was trained for glycopeptides and naked peptides, respectively, which indicates to some extent that ResNet machine learning method also has potential in solving missing value problem for standard proteomics.

The relevant discussions have been added in the discussion part as the reviewer suggested.

For-review Table 8 The Resnet machine learning method can improve missing values at PSM, peptide and proteins levels.

		cosine similarity		machine learning	
		glycopeptides	naked peptides	glycopeptides	naked peptides
PSM	PMVT	5.44%	8.98%	3.66%	2.46%
	PMVL	10.88%	17.92%	7.32%	4.90%
Peptide	PMVT	6.56%	10.12%	2.19%	1.43%
	PMVL	13.11%	20.20%	4.37%	2.83%
Protein	PMVT	6.43%	6.26%	3.22%	0.49%
	PMVL	12.87%	12.50%	6.43%	0.98%

Note: naked peptides presented the peptides without any glycosylation, here the naked peptides we used were identified by pFind software.

References

- [1] Cox, J. & Mann, M. MaxQuant enables high peptide identification rates, individualized p.p.b.-range mass accuracies and proteome-wide protein quantification. *Nat. Biotechnol.* **26**, 1367-1372 (2008).
- [2] Shen, X. et al. IonStar enables high-precision, low-missing-data proteomics quantification in large biological cohorts. *Proc. Natl. Acad. Sci. U. S. A.* **115**, E4767-E4776 (2018).
- [3] Lim, M.Y., Paulo, J.A. & Gygi, S.P. Evaluating False Transfer Rates from the Match-between-Runs Algorithm with a Two-Proteome Model. *J. Proteome Res.* **18**, 4020-4026 (2019).
- [4] Tyanova, S., Temu, T. & Cox, J. The MaxQuant computational platform for mass spectrometry-based shotgun proteomics. *Nat. Protoc.* **11**, 2301-2319 (2016).
- [5] K. He, X. Zhang, S. Ren and J. Sun, Deep Residual Learning for Image Recognition. *IEEE Conference on Computer Vision and Pattern Recognition (CVPR)* 770-778, (2016).
- [6] Zifeng Wu, Chunhua Shen, Anton van den Hengel, Wider or Deeper: Revisiting the ResNet Model for Visual Recognition. *Pattern Recogn.* **90**, 119-133 (2019).

- [7] Wu, N. et al. Deep Neural Networks Improve Radiologists' Performance in Breast Cancer Screening. *IEEE Trans. Med. Imaging* **39**, 1184-1194 (2020).
- [8] Chi, H. et al. Comprehensive identification of peptides in tandem mass spectra using an efficient open search engine. *Nat. Biotechnol.* **36**, 1059-1061 (2018).
- [9] Zeng, W.F., Cao, W.Q., Liu, M.Q., He, S.M. & Yang, P.Y. Precise, fast and comprehensive analysis of intact glycopeptides and modified glycans with pGlyco3. *Nat. Methods* **18**, 1515-1523 (2021).

5. 58% of glycopeptides (6435 out of 11001) quantified in more than two replicates for the HCC cell line experiment shows me that while pGlycoQuant may have improved missing value problems in quantitative glycoproteomics experiments, it has far from solved it. As such, I believe the authors must tone down the language that claims “could largely solve the problem of random precursor selection for MS2 analysis in different replicates.”

Reply:

Thanks for the reviewer’s comments. First of all, we want to explain that the HCC cell line experiment was based on the SILAC-labeling strategy, so the MBR, which is usually used in label-free strategy, was not adapted for the analysis among the four replicates in HCC cell line experiment. The result of “58% of glycopeptides (6435 out of 11001) quantified in more than two replicates” is mainly due to the 59.3% of the glycopeptides with more than duplicate identification among the four replicates (**For-review Figure 7**). If we only look at the missing values in each replicate, in which pGlycoQuant calculates the similarity based on the well-trained deep-learning-based evidence matching model between the light and heavy glycopeptides, less than 5% PMVL and 2% PMVT can be obtained (**For-review Table 9**).

Still, we agree that pGlycoQuant improved missing value problems in quantitative glycoproteomics experiments but has far from solve it. Correspondingly, we have modified the language as the reviewer suggested.

For-review Figure 7 The Venn Diagram of the glycopeptide identifications in the four replicates for the HCC cell line experiment. The 59.3% of the glycopeptides with more than duplicate identification among the four replicates were highlighted with red line.

For-review Table 9 The missing value analysis of quantitation results from each replicate in the HCC cell line experiment.

Group	Rep1	Rep2	Rep3	Rep4
PMVL (%)	4.31	4.51	4.15	4.62
PMVT (%)	1.62	1.63	1.48	1.75

6. How can site N979 change be relevant more so than the other four N-glycosites when the abundance change due to abundance differences at the protein level? It would seem all glycosites on L1CAM would have the same effect as N979 in the mutation experiments.

Reply:

Thanks for your question. Based on consideration of the following aspects, we selected the glycosite N979 as a candidate target for mutation experiments.

First, glycoprotein targets of FUT8 were enriched in cell migration proteins, including the adhesion molecule L1CAM, which has been demonstrated by published literatures [1-2]. Our data also showed that the FUT8 that regulates core fucosylation synthesis was changed in the HCC cell lines (**Supplementary Figure 23, Supplementary Data 6**), which implies that core fucosylation is highly correlated with

HCC cell metastasis. Among the 5 glycosites of L1CAM, site N979 was highly glycosylated compared with other four sites, especially, the core fucosylation with glycan Hex[5]HexNAc[4]NeuAc[1]Fuc[1] at glycosite 979 was significantly high in L1CAM in all three cell lines (**Supplementary Figure 24a**). Besides, all fucosylated glycans at site 979 of L1CAM were consistently upregulated with increasing metastatic potential of the cell lines (**Supplementary Figure 24b**).

Second, it has been reported that the cleavage of L1CAM by plasmin inhibits its ability to mediate cell invasion and metastatic outgrowth [3-6] and glycosylation of L1CAM can affect the access of plasmin to cleavage sites [1, 7]. Plasmin cleaves at two sites within the third FNIII domain following K842 or K845, resulting in a soluble fragment of 140 kDa and an intracellular fragment of 80 kDa [8-9]. The glycosite N979 and N849, which are closer to the cleavage domain than the other glycosites [3, 10], are more likely to be relevant for maintaining the stability of the structure and functions of L1CAM. While, the proportion of glycosylation at site N849 is not high (**Supplementary Figure 24a**) and the main glycan type is oligomannose rather than core fucosylation (**Figure 6a**).

Third, the results in the manuscript showed that the changed fucosylation abundance of L1CAM N979 was not only due to the L1CAM protein level itself but also due to the FUT8 level (**Figure 6c-f**): from no metastatic potential to low metastatic potential, the increased fucosylation at site 979 was due to the increased expression of FUT8; from low metastatic potential to high metastatic potential, the increased fucosylation at site 979 was mainly due to the increased protein content of L1CAM. A comparison of differential intact glycopeptides without and with normalization to protein abundance also illustrated that (**Supplementary Figure 25, Supplementary Data 7**).

Taking the above aspects into consideration, we think glycosite N979 is more relevant to the HCC metastasis than the other four sites on L1CAM. However, since no molecular biology experiment was carried out on the other four sites in this study, we cannot get a conclusion whether the other four sites have the same effect or not. While, we think the reviewer raised an interesting question that is whether changes in glycoprotein level or in site-specific glycosylation level are more relevant to the function of a glycoprotein, which is not addressed in this work and deserves further investigation.

References:

- [1] Agrawal, P. et al. A Systems Biology Approach Identifies FUT8 as a Driver of Melanoma Metastasis. *Cancer Cell* **31**, 804-819.e7 (2017).
- [2] Yu, M. et al. FUT8 drives the proliferation and invasion of trophoblastic cells via IGF-1/IGF-1R signaling pathway. *Placenta* **75**, 45-53 (2019).
- [3] Maten, M.V., Reijnen, C., Pijnenborg, J.M.A. & Zegers, M.M. L1 Cell Adhesion Molecule in Cancer, a Systematic Review on Domain-Specific Functions. *Int. J. Mol. Sci.* **20**, 4180 (2019).
- [4] Yu, X., Yang, F., Fu, D.L. & Jin, C. L1 cell adhesion molecule as a therapeutic target in cancer. *Expert Rev. Anticancer Ther.* **16**, 359-371 (2016).
- [5] Min, J.K. et al. L1 cell adhesion molecule is a novel therapeutic target in intrahepatic cholangiocarcinoma. *Clin. Cancer Res.* **16**, 3571-3580 (2010).
- [6] Kiefel, H. et al. L1CAM: a major driver for tumor cell invasion and motility. *Cell Adh. Migr.* **6**, 374-384 (2012).
- [7] Yang, D. et al. Increased plasmin-mediated proteolysis of L1CAM in a mouse model of idiopathic normal pressure hydrocephalus. *Proc. Natl. Acad. Sci. U. S. A.* **118**, e2010528118 (2021).
- [8] Silletti, S., Mei, F., Sheppard, D. & Montgomery, A.M. Plasmin-sensitive dibasic sequences in the third fibronectin-like domain of L1-cell adhesion molecule (CAM) facilitate homomultimerization and concomitant integrin recruitment. *J. Cell Biol.* **149**, 1485-1502 (2000).
- [9] Li, Y. & Galileo, D.S. Soluble L1CAM promotes breast cancer cell adhesion and migration in vitro, but not invasion. *Cancer Cell Int.* **10**, 34 (2010).
- [10] Colombo, F. & Meldolesi, J. L1-CAM and N-CAM: From Adhesion Proteins to Pharmacological Targets. *Trends Pharmacol. Sci.* **36**, 769-781 (2015).

7. Can the authors provide annotated spectra for the 2% of glycopeptide IDs that had the N-X-C motif? That number is close to the FDR cutoff, and it is within reason to believe they may not be accurate identifications.

Reply:

Thanks for your careful review. All the annotated spectra for the 92 glycopeptides with N-X-C motif were provided in the **For-review Data 3**. After we manually checked these spectra, except for 22 glycopeptides with multiple glycosylation sites that could not be 100% confirmed as the N-X-C motif, the others clearly indicated the

glycosylation site with N-X-C motif. Some of the annotated spectra as examples were also shown as below (**For-review Figure 8**). Actually, other studies have also reported N-glycosylation with N-X-C motif [1-8]. For example, Matthias Mann et al. reported rarely N-glycosylation with N-X-C motif in mouse tissues (about 1.28%) [7]. Hao Yang et al found almost 2.3% glycosites with the N-X-C motif in human plasma [8].

References:

- [1] Gámez, G. et al. Atypical N-glycosylation of SARS-CoV-2 impairs the efficient binding of Spike-RBM to the human-host receptor hACE2. *bioRxiv*, DOI:10.1101/2021.04.09.439154 (2021).
- [2] Fichelova, V. et al. Functional identification of potential non-canonical N-glycosylation sites within Ca(v)3.2 T-type calcium channels. *Mol. Brain* **13**, 149 (2020).
- [3] Dang, L. et al. Mapping human N-linked glycoproteins and glycosylation sites using mass spectrometry. *Trends Anal. Chem.* **114**, 143-150 (2019).
- [4] Yang, G. et al. Comprehensive Glycoproteomic Analysis of Chinese Hamster Ovary Cells. *Anal. Chem.* **90**, 14294-14302 (2018).
- [5] Yasuda, D., Imura, Y., Ishii, S., Shimizu, T. & Nakamura, M. The atypical N-glycosylation motif, Asn-Cys-Cys, in human GPR109A is required for normal cell surface expression and intracellular signaling. *FASEB J.* **29**, 2412-2422 (2015).
- [6] Sun, S. & Zhang, H. Identification and Validation of Atypical N-Glycosylation Sites. *Anal. Chem.* **87**, 11948-11951 (2015).
- [7] Zielinska, D.F., Gnad, F., Wiśniewski, J.R. & Mann, M. Precision mapping of an in vivo N-glycoproteome reveals rigid topological and sequence constraints. *Cell* **141**, 897-907 (2010).
- [8] Zhang, Y. et al. Glyco-CPLL: An Integrated Method for In-Depth and Comprehensive N-Glycoproteome Profiling of Human Plasma. *J. Proteome Res.* **19**, 655-666 (2020).

8. The match in runs feature is interesting and strengthens the manuscript. However, it is the algorithm within pGlycoQuant that has the most vulnerability to false matches. The sialic acid tests they performed produced believable data that warrants inclusion in the manuscript, but these are still highly contrived experiments. I remain cautious that this would prove reliable in a sample with thousands of sialylated glycopeptides with its current implementation. Thus, I think the authors should state this even more explicitly than they have in the discussion, saying that Match In Runs is untested on truly complex mixtures and should be thoroughly evaluated manually any time it is used.

Reply:

Thanks for your suggestion. We have added the corresponding contents in the discussion part as suggested.

9. In the discussion, the authors should suggest the best tools to use in tandem with pGlycoQuant to manually inspect their data, as they have done throughout. Is it with a tool like Skyline? How can we in the rest of the field evaluate our data after we have used pGlycoQuant so that we can check the same quality metrics in our own data that the authors have reported here? This includes MS1 glycopeptide traces/isotope patterns and MS2 spectra.

Reply:

In the pGlycoQuant software package, there is a separate application called UltraVisual, which is an XIC curve extraction tool supporting the construction of the chromatographic curves through input the quantitative results and ms1 data files. The UltraVisual can help to intuitively check whether the extracted signals for quantitation are reliable or not. The following **For-review Figure 9a** is the screenshot of the UltraVisual interface. After the ini files (will be filled automatically), ms1 data files, quantitative results (pGlycoQuant.spectra.list) and the output path are filled up correctly (as shown in the blue frames) in the UltraVisual interface (**For-review Figure 9a**) and “run” button is pushed, the corresponding reconstructed glycopeptide chromatographic curves can be outputted (**For-review Figure 9b**). The UltraVisual tool and the corresponding detailed usage method, which have been included in the UltraVisual folder in the pGlycoQuant software tool package, are freely available (see “Code availability” and “Information for reviewers” in the manuscript for the download information).

For-review Figure 9 The usage of UltraVisual for the view of the XIC curve. (a) The screenshot of the UltraVisual interface and the usage. (b) The XIC plots of “1902-Hela-Labelfree” dataset generated by UltraVisual.

10. Can the authors expand on the rare glycopeptides quantified in the deglycoproteomics experiment? Are they are because they are glycopeptides whose glycans would not be expected to be removed by PNGaseF? Reporting these glycopeptides and perhaps some annotated spectra would be helpful to understand what makes them rare.

Reply:

Thanks for the reviewer’s careful review. A total of 3 and 31 glycopeptides were quantified in the deglycoproteomics experiment for fission yeast (**For-review Table 10**) and human serum (**For-review Table 11**), respectively. None glycopeptides were quantified in the deglycoproteomic experiment for standard glycoprotein IgG. The original quantitation data can also be obtained from the MassIVE (<https://massive.ucsd.edu/>) with identifier MSV000089484 (see the Data availability in the manuscript for the detailed download information). The corresponding annotated spectra for these glycopeptides were included in the **For-review Data 4**, some of which were also shown below (**For-review Figure 10**). We think that the main reason for the rare glycopeptides quantified in the deglycoproteomics experiment is that PNGase F is

not 100% efficient at releasing glycans in complex samples [1]. In general, the efficiency of PNGase F on removal of glycans from single glycoproteins is up to 100% [2], which was also supported by our results from the standard glycoprotein IgG (**Supplementary Figure 5e**). However, in complex samples, PNGase F may randomly fail to recognize the innermost region between the GlcNAc and asparagine residues due to the great variety of glycopeptides and complex glycan structures in the complex samples, especially for those glycopeptides with large glycan structures, resulting in some glycopeptides without efficient removal of glycans [3-5]. As we can see in the **For-review Table 10**, all the glycans of the three glycopeptides from the fission yeast are high-mannose type with more than 10 glycan units. Among the 31 glycopeptides from the human serum, three glycopeptides contain high-mannose glycan type, twenty-eight glycopeptides are complex/hybrid glycan type (**For-review Figure 11a**), and about 77% of them contain the glycan with more than 10 glycan units (**For-review Figure 11b**). Most of these un-removal glycopeptides have large glycans, which may affect the effective glycan releasing of PNGase F. The above relevant contents have been added in the Supplementary Figure 5.

References:

- [1] Kita, Y. et al. Quantitative Glycomics of Human Whole Serum Glycoproteins Based on the Standardized Protocol for Liberating N-Glycans. *Mol. Cell. Proteomics* **6**, 1437-1445 (2007).
- [2] Mann, A.C., Self, C.H. & Turner, G.A. A general method for the complete deglycosylation of a wide variety of serum glycoproteins using peptide-N-glycosidase-F. *Glycosylation Dis.* **1**, 253-261 (1994).
- [3] Maley, F., Trimble, R.B., Tarentino, A.L. & Plummer, T.H., Jr. Characterization of glycoproteins and their associated oligosaccharides through the use of endoglycosidases. *Anal. Biochem.* **180**, 195-204 (1989).
- [4] Plummer, T.H., Jr. & Tarentino, A.L. Purification of the oligosaccharide-cleaving enzymes of *Flavobacterium meningosepticum*. *Glycobiology* **1**, 257-263 (1991).
- [5] Tretter, V., Altmann, F. & März, L. Peptide-N4-(N-acetyl-beta-glucosaminyl) asparagine amidase F cannot release glycans with fucose attached alpha 1----3 to the asparagine-linked N-acetylglucosamine residue. *Eur. J. Biochem.* **199**, 647-652 (1991).

For-review Table 10 The glycopeptides quantified in the deglycoproteomics experiment for fission yeast.

Protein	Site	Glycan(H,N,A,F)	Glycan size
sp O94559 YF65_SCHPO	232	[9 2 0 0]	11
sp Q9USU5 GAS2_SCHPO	344	[18 2 0 0]	20
sp Q9USU5 GAS2_SCHPO	344	[17 2 0 0]	19

For-review Table 11 The glycopeptides quantified in the deglycoproteomics experiment for human serum

Protein	Site	Glycan(H,N,A,F)	Glycan size
sp O95045 UPP2_HUMAN	9	[5 4 1 2]	12
sp O95445 APOM_HUMAN	135	[5 3 1 0]	9
sp P00450 CERU_HUMAN	138	[5 4 1 0]	10
sp P00738 HPT_HUMAN	241	[5 4 1 0]	10
sp P00738 HPT_HUMAN	241	[5 7 1 2]	15
sp P00738 HPT_HUMAN;sp P00739 HPTR_HUMAN	184;126	[5 4 2 0]	11
sp P01031 CO5_HUMAN	741	[6 5 3 0]	14
sp P01859 IGHG2_HUMAN	176	[4 4 0 1]	9
sp P01871 IGHM_HUMAN	46	[5 4 1 0]	10
sp P01871 IGHM_HUMAN	46	[6 3 1 0]	10
sp P01876 IGHA1_HUMAN	340	[5 2 0 0]	7
sp P01876 IGHA1_HUMAN	340	[5 5 0 0]	10
sp P01877 IGHA2_HUMAN;sp P0DOX2 IGA2_HUMAN;sp P01876 IGHA1_HUMAN	131;246;144	[5 4 2 0]	11
sp P02763 A1AG1_HUMAN	103	[7 5 2 1]	15
sp P02763 A1AG1_HUMAN	103	[6 5 3 0]	14
sp P02763 A1AG1_HUMAN;sp P19652 A1AG2_HUMAN	56;56	[5 4 1 0]	10
sp P02763 A1AG1_HUMAN;sp P19652 A1AG2_HUMAN	56;56	[6 5 3 1]	15

sp P02763 A1AG1_HUMAN;sp P19652 A1AG2_HUMAN	56;56	[5 4 2 0]	11
sp P02763 A1AG1_HUMAN;sp P19652 A1AG2_HUMAN	56;56	[7 5 1 1]	14
sp P02763 A1AG1_HUMAN;sp P19652 A1AG2_HUMAN	56;56	[6 5 2 1]	14
sp P02763 A1AG1_HUMAN;sp P19652 A1AG2_HUMAN	56;56	[6 5 2 0]	13
sp P02763 A1AG1_HUMAN;sp P19652 A1AG2_HUMAN	56;56	[6 5 3 0]	13
sp P02763 A1AG1_HUMAN;sp P19652 A1AG2_HUMAN	56;56	[7 5 2 1]	15
sp P02787 TRFE_HUMAN	630	[6 5 2 0]	13
sp P04004 VTNC_HUMAN	242	[6 5 3 1]	15
sp P05546 HEP2_HUMAN	188	[5 4 1 0]	10
sp P0DOX5 IGG1_HUMAN;sp P01857 IGHG1_HUMAN	299;180	[4 4 0 1]	9
sp P0DOX5 IGG1_HUMAN;sp P01857 IGHG1_HUMAN	299;180	[5 4 0 1]	10
sp P0DOX5 IGG1_HUMAN;sp P01857 IGHG1_HUMAN	299;180	[3 4 0 1]	8
sp Q7RTR8 T2R42_HUMAN	304	[3 2 0 0]	5
sp Q9ULR3 PPM1H_HUMAN	354	[4 2 0 0]	6

For-review **Figure 10** Annotated tandem mass spectra of the glycopeptides from fission yeast (a) and human serum (b) as examples.

For-review Figure 11 (a) The distribution of the glycopeptides containing high mannose type, complex type and hybrid type quantified in the deglycoproteomic experiment for human serum grouped by glycan size. (b) The glycan size distribution in 31 glycopeptides from human serum.

REVIEWERS' COMMENTS

Reviewer #1 (Remarks to the Author):

The authors presented another revision of the manuscript entitled as "pGlycoQuant with a deep residual network ...". The reviewer really appreciates the amount of work but some queries have not received a clear answer.

Below you can find my comments:

1) To the query of what is special between the quantification of peptides and glycopeptides authors did not provide a clear answer. They stated again that "...the challenge is even bigger in glycopeptide quantification...". But what is such a challenge? Is it because glycopeptides are low abundant and peptides dominate in the mixture? Is it because glycopeptides are masked by the chemical noise? Is it because glycopeptide demonstrates special isotopic distribution? The authors demonstrated a machine-learning approach to cluster between peptides and glycopeptides. This is acceptable. But why is it so? The review could guess that this is because the authors selected N-glycopeptide mixture for the analysis, where glycan part is large and contribute more to the net isotopic distribution. Average glycan and average peptide (with similar masses) indeed could have different C13 distributions. Therefore isotopic pattern is slightly different and could be recognized? What about glycopeptide with the short glycan part, where there is no such difference? Could it be then that the software is limited only to N-glycopeptides? This should be properly discussed.
All other queries were properly addressed

Reviewer #3 (Remarks to the Author):

The authors have done commendable work to address my questions and comments. I hope they can see the strengths they have added to their manuscript by thoughtfully and thoroughly considering concerns from all reviewers. I have no further comments and see this manuscript as suitable for publication.

Point-by-point response to the reviewer's comments

pGlycoQuant with a deep residual network for quantitative glycoproteomics at intact glycopeptide level enabling the functional exploration of site-specific glycosylation

Reviewers' Comment

Reviewer #1:

The authors presented another revision of the manuscript entitled as "pGlycoQuant with a deep residual network ...". The reviewer really appreciates the amount of work but some queries have not received a clear answer.

Below you can find my comments:

1) To the query of what is special between the quantification of peptides and glycopeptides authors did not provide a clear answer. They stated again that "...the challenge is even bigger in glycopeptide quantification...". But what is such a challenge? Is it because glycopeptides are low abundant and peptides dominate in the mixture? Is it because glycopeptides are masked by the chemical noise? Is it because glycopeptide demonstrates special isotopic distribution? The authors demonstrated a machine-learning approach to cluster between peptides and glycopeptides. This is acceptable. But why is it so? The review could guess that this is because the authors selected N-glycopeptide mixture for the analysis, where glycan part is large and contribute more to the net isotopic distribution. Average glycan and average peptide (with similar masses) indeed could have different C13 distributions. Therefore isotopic pattern is slightly different and could be recognized? What about glycopeptide with the short glycan part, where there is no such difference? Could it be then that the software is limited only to N-glycopeptides? This should be properly discussed.

All other queries were properly addressed

Reply:

In the last revision, we have compared the value distribution of the 1024×1 vector that were transformed from the evidence as a matrix between the naked peptides and glycopeptides. The comparison results showed that glycopeptides and naked peptides had different feature distributions and the features of glycopeptides presented more significant fluctuations than that of naked peptides (**For-review Figure 1 in the last revision, also shown below**). This comparison and, together with other analyses in the

last revision, e.g., the comprehensive comparison of pGlycoQuant with several widely used proteome software tools showing outperformance of pGlycoQuant in the aspect of both reducing missing values and quantitation precision, and the dedicatedly designed experiment for the quantitation of glycopeptides with decreased concentrations demonstrating the higher sensitivity of pGlycoQuant in detecting low-abundance glycopeptide signals, illustrated from the side that the overall differences exist between the quantification of peptides and glycopeptides and suggested the necessity of developing dedicated software for intact glycopeptide quantitation.

In this revision, we made further exploration of the individual features. The results showed that the glycopeptides masked by naked peptides poses challenge to glycopeptide quantification (**For-review Figure 2**), and the analysis of some classical features used in the machine-learning approach indicates what is special between the quantification of peptides and glycopeptides (**For-review Figure 3**). Briefly, we quantified the naked peptides and the glycopeptides in HeLa cell samples that were analyzed by LC-MS/MS after tryptic digestion without glycopeptide enrichment or with glycopeptide enrichment, respectively. It was shown that naked peptides legitimately dominated in the non-enrichment samples (**For-review Figure 2a**) and also covered over the glycopeptides though after glycopeptide enrichment (**For-review Figure 2b**), which presents a great challenge for glycopeptide quantification to find the target glycopeptide evidence from the interferences. For the isotopic distribution, although the theoretical isotopic peak distribution of the naked peptides and the glycopeptides with similar mass (tolerance within 20 ppm) have high similarities with cosine similarity of more than 0.99 (**For-review Figure 4**), the similarity of the isotopic chromatograms between glycopeptides and peptides is not high, which could be illustrated by our further analyses on the performance of machine learning NP and machine learning GP on six classical features in the glycopeptides quantitation (**For-review Figure 3**).

As we have shown in the **For-review Figure 1** in the last revision (also shown below), multiple different feature distributions exist in glycopeptide and peptide quantification. The deep learning (DL) algorithm in the pGlycoQuant does not just use individual features. It is a general algorithm, which can train different models to adapt to glycopeptide or peptide quantification. Algorithmically, DL-based pGlycoQuant can learn similarities and differences no matter whether glycopeptides contain long or short glycans (we briefly showed the application of pGlycoQuant to O-glycopeptide

quantification in human serum and got the fine results as shown **For-review Figure 5**), or even naked peptides, which is exactly the advantage DL brings for quantification. Some of the corresponding discussions have already been added in the Discussion part of the manuscript. In terms of DL-based models that were trained from peptides and glycopeptides, they are indeed different and thus dedicated model-based tool for glycopeptide quantification is necessary, which has been already demonstrated in the last revision (**For-review Table 1 in the last revision, also shown below**) and also further demonstrated in this revision as shown in the **For-review Figure 3**.

At last, we want to thank the reviewer for his/her insightful concerns, which prompt our thought. In search of the exact differences between glycopeptide and naked peptide quantification would help us design a deterministic algorithm and combine with DL to improve the interpretability of the algorithm and perhaps the quantitation performs.

For-review Figure 1 The value distribution of inputted 1024×1 vector. The x axis represents the 1024×1 vector, and the y axis represents the value of vector. (a) Distribution of 1024 points of 1024×1 vector in default order. (b) Grading the vector of naked peptides in descending order.

For-review Figure 2 The intensity distribution of the naked peptides and the glycopeptides in HeLa cell samples that were analyzed by LC-MS/MS after tryptic digestion without glycopeptide enrichment (a) or with glycopeptide enrichment (b).

For-review Figure 3 The performance of machine learning NP and machine learning GP on six classical features (Supplementary Figure 2) in the glycopeptide quantitation. The machine learning NP and the machine learning GP were used for the quantitation of each identified glycopeptide. The machine learning NP refers to the machine learning model that was trained by naked peptides and the machine learning GP refers to the machine learning model that was trained by glycopeptides. The score distribution was displayed on six classic features selected in machine learning-based evidence matching model, among which, the closer the score of the “Distance of retention time” and the “Intensity difference” is to 0 and the score of other features is to 1, the better the results are. The features of “Similarity of mono isotopic chromatograms”, “Similarity of mono+1 isotopic chromatograms”, and “Similarity of mono+2 isotopic chromatograms” represented the similarity of chromatograms of mono, mono+1 and mono+2 isotopic peaks of the evidences from two runs respectively. One glycopeptide performed the same chromatography profiles in two runs (i.e. the “Similarity of mono isotopic chromatograms” and the other two features are 1). The feature of “Similarity of environment matrix” represented the similarity of all signals (considered as one big matrix) in the entire whole retention time window from two runs. This feature took into account contributions from multiple peptide evidences and other random co-eluting species in the entire retention time. As in complex samples the true evidence is usually accompanied by other candidates in the retention time window (i.e. the “Similarity of

environment matrix” is not 1), it is difficult to match the evidence correctly in MBR. The feature of “Distance of retention time” represented the distance of retention time of the evidences between two runs.

The figure showed that compared with the use of machine learning NP, the feature score distribution from the glycopeptide quantitative results improved when using the machine learning GP, which indicated that the two models trained respectively are different since feature differences exist between naked peptides and glycopeptides. The differences of these features are a potential answer to the query of what is special between the quantification of peptides and glycopeptide.

For-review Figure 4 The similarity of the isotopic peak distribution between the naked peptides and glycopeptides, whose precursor masses were within 20 ppm.

For-review Figure 5 The evaluation on label-free quantitation of O-glycopeptides in human serum with triplicates of LC-MS/MS. (a) The missing value analysis of quantitative intensity of O-glycopeptide data (including proportion of missing value in line (PMVL) and proportion of missing value in total (PMVT)). (b) The correlation coefficient of quantitative intensity of O-glycopeptide data. The lower left part is the scatter distribution diagram of intensity of light glycopeptide and heavy glycopeptide, and the upper right part is the calculated Pearson linear correlation coefficient.

For-review Table 1 The performance of machine learning GP and machine learning NP on glycopeptide quantitation in terms of reducing missing values and quantification correlation. As the results shown in this table, compared with the machine learning NP method, the machine learning GP method has better quantification performance in terms of reducing missing values and quantification correlation, which indicates to some extent that feature differences exist between naked peptides and glycopeptides and the quantification tool dedicated for glycopeptides is needed.

		machine learning GP for glycopeptide quantitation	machine learning NP for glycopeptide quantitation
PSM	PMVT	3.66%	17.74%
	PMVL	7.32%	35.48%
	Pearson	0.992	0.801
	Std.	0.145	1.176
Peptide	PMVT	2.19%	9.00%
	PMVL	4.37%	18.01%
	Pearson	0.989	0.960
	Std.	0.178	0.558
Protein	PMVT	3.22%	7.89%
	PMVL	6.43%	15.89%
	Pearson	0.991	0.973
	Std.	0.16	0.395

Note: machine learning NP refers to the machine learning model that was trained by naked peptides and machine learning GP refers to the machine learning model that was trained by glycopeptides.

Reviewer #3:

The authors have done commendable work to address my questions and comments. I hope they can see the strengths they have added to their manuscript by thoughtfully and thoroughly considering concerns from all reviewers. I have no further comments and see this manuscript as suitable for publication.

Reply:

We thank the reviewer again for the approval of pGlycoQuant. We also appreciate all the reviewers' valuable comments and suggestions, which greatly improved our work after fixing them, and we have learned a lot from the revision processes.